## Registered report

psychology

disclosure, intelligence gathering, intelligence interviewing, pragmatic inference, relevance theory

**Author for correspondence:**
David A. Neequaye
e-mail: daneequaye@gmail.com

# How intelligence interviewees mentally identify relevant information

David A. Neequaye[1] and Alexandra Lorson[2]

[1]Department of Psychology, University of Gothenburg, Gothenburg 40530, Sweden
[2]School of Philosophy, Psychology and Language Sciences, University of Edinburgh, Edinburgh, UK

 DAN, 0000-0002-7355-2784

This research explored how intelligence interviewees mentally identify the relevant information at their disposal. We theorized that interviewees estimate the interviewer's objectives based on how they frame any attempt to solicit information. Then interviewees organize the information they possess into item designations that pragmatically correspond to the perceived interviewer-objective. The more an interviewer specifies what they want to know, the more the interviewee will mentally designate information items corresponding with that objective. To examine the theory, we conducted two identical experiments wherein participants assumed the role of an informant with one of two dispositions. They were to be cooperative or resistant when undergoing an interview. The interviewer posed specific or ambiguous questions. In Study 1 ($N = 210$), interviewees identified applicable information items based on their interviewer's questions. And interviewees answered their interviewer's questions in Study 2 ($N = 199$). We aimed to demonstrate that question type influences mental designations and disposition affects disclosures. Disposition had a stronger influence on interviewees' disclosure than when reasoning about what the interviewer wants to know. But contrary to our expectations, mental designation preferences indicated that interviewees generally assume interviewers want to know complete details, irrespective of question specificity. We suggest avenues for future research.

## 1. Introduction

In human intelligence interviews, interviewers solicit information from interviewees for national or international security purposes

(e.g. [1]). This research examines how such interviewees mentally identify or designate applicable items of information in interviews.[1]

The thesis of this article is that during an interview, interviewees mentally designate applicable items of information. Designations here refer to an interviewee's mental representation of a coherent unit of information. That is, information that, if the interviewee desires, they can make the interviewer aware of, or ideally communicate clearly and logically.[2] Interviewees may consider such designations to be different or separate from other similarly designated information items. However, different information items need not be mutually exclusive. Interviewees can view information items as separate but related units when events induce links between the items. We use the word—applicable— to indicate the following proposition. The designations an interviewee mentally demarcates will inevitably be limited to the subject of a prospective or ongoing interview, as opposed to the interviewee's general knowledge. Later, we explain in detail how interviewees *mentally* determine applicable information items.

An illustration may assist in understanding the postulations just described. Consider the following fictitious scenario. An informant has discovered that a criminal network *smuggles oxycodone into a prison using ambulances*. The informant will be questioned in a routine meeting. Let us assume that the criminal network is the usual reason for the meetings. Thus, the interviewer knows about the network but not the informant's recent discovery. We contend that the informant's discovery and, by extension, many subjects of interest could be mentally represented as a single information unit or multiple items—with varying degrees of completeness. Here, completeness refers to the full extent of an interviewee's knowledge on the subject. In the current example, the informant could mentally represent the discovery in any or all of the following formats. She could truly think that the interviewer might want to know any of the following things about the criminal network.

(z1) They smuggle oxycodone into a prison using ambulances.
(z2) They smuggle oxycodone into a prison.
(z3) They smuggle oxycodone.
(z4) They smuggle oxycodone using ambulances.

We use the preceding sentences to communicate the possible mental representations, in the present illustration, to the reader. In an actual instance, information units need not be or be thought of as complete sentences. We are positing that such representations—whatever their format or completeness—will embody what an interviewee envisions they *could* disclose. The word *could* is not trivial here. The interviewee may disclose or withhold the mentally identified information depending on further strategic considerations. Mentally designating information items is a crucial precursor to disclosure considerations and decisions, but this aspect of interviewees' cognitive process remains unexamined. This article aims to fill the flagged research gap.

In the current intelligence interviewing research paradigm, researchers motivate participants to disclose and withhold information (e.g. [2–4]). Participants who assume the role of mock interviewees receive a twofold instruction. They are instructed not to share too little information because assisting the interviewer may be beneficial. They are also told not to share too much information to avoid risks of disclosure. These instructions allow research studies to mimic the typical dilemma interviewees face in real intelligence interviews (e.g. [5]). A recent review indicates that when responding to direct questions, the mock interviewee role leads participants to disclose some but not all the information they possess [6]. We can infer from this finding that interviewees implement a *honing* process, at least intuitively, to determine the relevant information at their disposal. And they disclose their preferred items out of the lot. Put differently, interviewees mentally designate applicable information items and subsequently choose what to disclose.

The processes by which interviewees mentally itemize what they could disclose remain unknown. As a reminder, here, we do *not* mean the information an interviewee is necessarily willing to disclose or chooses to disclose. We mean: how an interviewee determines the germane information at their disposal. Subsequently, depending on the circumstances at hand, an interviewee may fully disclose or

---

[1]Housekeeping notes: the phrases items of information and units of information are used synonymously throughout this article. We deviate from the standard presentation style of empirical articles. The present article is a hybrid that first presents a nascent theory and then proposes initial studies to examine the theory's core premise. We advise the reader to expect a more in-depth introduction than usual before the empirical aspect of the article.

[2]As in everyday communication, clarity can be subjective. What an interviewee considers clear and logical may not be perceived by another as such.

withhold the information altogether. One could also reveal partial bits of the information. Or the interviewee could replace the information with a false one and deceive the interviewer. Most research studies focus on these behaviours that interviewees enact after mentally designating applicable information items. No research endeavour has specified the mechanisms by which interviewees mentally designate applicable information units in the first place. To our knowledge, there is currently no concerted effort in that regard.

Understanding how interviewees mentally establish information item designations will expand current insights about the underpinnings of disclosure in intelligence interviews. Such knowledge allows the possibility to scrutinize another layer of influence: how interviewing approaches also impact the mental designation of information items to affect disclosure. This insight will add to existing research examining the processes that affect disclosure. In short, this article explores how interviewees mentally determine the germane information at their disposal, which they may or may not disclose.

# 2. Deciphering information objectives and the consequences thereof

Empirical and anecdotal evidence indicates that unyielding interviewees in investigative interviews employ various countermeasures to avoid cooperating with interviewers [5,7,8]. For example, sometimes, such interviewees claim forgetfulness. We can thus assume that generally, unyielding interviewees intuitively estimate the subject of interest or an interviewer's information objectives to evade fully cooperating with such interests. By extension, it also stands to reason that cooperative interviewees similarly determine what the subject of interest is before cooperating by providing the commensurate useful information. We contend that, in intelligence interviews, like typical investigative interviews, interviewees try to understand what subjects are of interest to a prospective or current interviewer. From such understandings, interviewees *mentally hone in* on what information out of their general knowledge may be applicable.

## 2.1. Relevance theory as an account of construing information objectives

Theorists in pragmatics have established that individuals decode communicators' utterances to understand the possible messages being relayed [9,10]. Consequently, a sender's utterances, whether verbal or non-verbal, serve as inputs guiding the receiver's comprehension. According to relevance theory, such inputs gain significance or *relevance* based on a receiver's evaluation that the input will contribute a worthwhile difference to their comprehension of the world [11]. These useful comprehensions include: reinforcing, modifying or discarding prior understandings. More relevant inputs in that regard are likelier to be processed to a deeper degree—they will exert higher levels of influence on a receiver's comprehension [12]. Importantly, individuals actively search for relevant inputs when deciphering a sender's meaning. Those inputs are typically interpreted within the context of the supposed interaction. The specific circumstances at hand contribute to the saliency of relevant inputs such that some inputs become more salient than others in different settings [12].

The following illustration demonstrates the tenets of relevance theory just described. Imagine that you intend to watch a television show, *Fun Psyc*, scheduled to air at 19.00 on Channel UMO. At 18.55, you turn on your TV and see that the station is reporting an event they claim is breaking news: a highway motor accident. The input you have just received may lead you to conclude that Fun Psyc could be delayed. You may have to wait just a little longer, 5 min. Now consider the addition of further contextual information. For instance, the knowledge that Channel UMO's coverage of highway motor accidents usually lasts approximately one hour. Such contextual information may lead the input to gain relevance yielding further implications. For example, that Fun Psyc could be postponed, and you are better off making other plans.

We can think of an interviewer's solicitation attempts as information-seeking utterances or inputs. These inputs gain relevance based on an interviewee's estimation that an utterance contributes a worthwhile difference to understanding what the interviewer wants to know.[3] Consequently, more relevant inputs exert greater influence on an interviewee's comprehension of the objectives at hand.

---

[3]We use the terms *information-seeking utterance* and *input* in a broad sense. The terms include all the potential ways an interviewer may deliberately or inadvertently communicate their information objectives. Thus, solicitation attempts may include clear inquiries such as direct questions, vague solicitations such as a general request for an interview, or statements intended to influence the interviewee to make an inference about a supposed question.

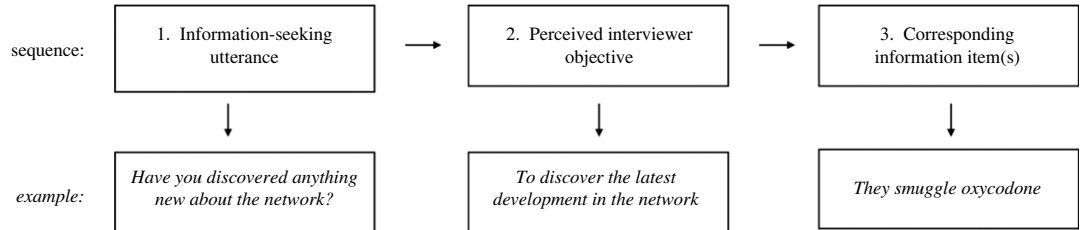

**Figure 1.** The proposed sequence by which interviewees subjectively designate information items.

Interviewees then mentally organize the information they possess into item designations they believe correspond to the objective of those relevant information-seeking utterances. Recall the earlier example involving informing on a criminal network: mentally, the informant can *subjectively* determine the following. Out of the information possessed, item z3—they smuggle oxycodone—best matches the interviewer's perceived objective to discover the latest development in the network. The objective could be deciphered from the information-seeking utterance—have you discovered anything new about the network? Figure 1 illustrates the proposed sequence by which interviewees subjectively designate information units mentally.

We contend that such systematization is a necessary first step, which subsequently undergoes any further processing that influences disclosure of the information. An example of such further processing could be managing the expected outcomes of disclosure by strategically sharing items that are ostensibly beneficial or innocuous to one's self-interests [13]. Similar to navigating many social interactions, understanding information objectives and mentally demarcating corresponding information units may recur throughout an interview. The interviewee stops mentally organizing when they exhaust the interviewer's information-seeking utterances with possible matching information items. The process restarts when the interviewee perceives novel utterances that require matching with commensurate information units. Such novel utterances include an interviewee's reinterpretation of existing utterances.

The reader should note that our use of the verb *to organize*, in the preceding paragraph, is to emphasize the following stipulation. Interviewees implicitly or explicitly try to make sense of the information they possess in relation to whatever they perceive an interviewer's information objectives to be. That is, the applicability of the information in the interview. An implicit attempt at determining such applicability could be forming a hypothesis about the interviewer's objectives. Then the interviewee might privately itemize the information that may fulfil those objectives. An explicit attempt could be asking the interviewer what their goals are. After such an implicit or explicit determination of what the interviewee considers the germane information, the interviewee may perform any of the following behaviors. They may disclose the appropriate information, provide incomplete information, decline to assist or lie. We are not suggesting that the interviewee's information is necessarily *unorganized* in the sense that it is meaningless prior to an interview. Indeed, such information can be substantive to an interviewee in contexts apart from an interview—for example, when discussing the subject with a friend.

# 3. Ensuing predictions

Exponents of relevance theory have noted that a receiver's deciphering of a sender's meaning is predominantly a non-demonstrative process [12]. Nevertheless, the natural results of such a process may provide opportunities to examine the mechanism's verisimilitude. Thus, the current research needs to offer testable predictions about the following question. Can the mental designation of information items be differentially manipulated by varying the characteristics of information-seeking utterances? Next, we explore such possibilities.

## 3.1. The clarity and worthwhileness of information-seeking utterances

It is well established that humans have limited cognitive resources. Hence, we are inclined to manage those resources most efficiently when making sense of stimuli in our environment [14,15]. In this view, relevance theory posits that we actively seek out *clear* and *worthwhile* inputs that we estimate

will contribute to understanding a sender's message. Conversely, receivers usually avoid discerning senders' meanings using messages that they perceive as convoluted and provide little insight about what a sender means [11]. As such, utterances that one perceives to be clearer and more worthwhile are likelier to determine the meaning a receiver derives from the messages a sender transmits (e.g. [16]). Consider the following illustration modelled after one described by Wilson & Sperber [12, p. 252]. Jemima wants to ask Maria if Maria is interested in having lunch together. Jemima could truthfully communicate the request in any of the following ways.

1. Do you want to have lunch with me?
2. If drinking too much water is fatal, I am asking if you want to have lunch with me.

*Utterance-1 and utterance-2* are similar in one important sense: they both contain Jemima's request. Nonetheless, utterance-1 is arguably more comprehensible; it provides immediate insight into Jemima's request, and Maria requires little effort to understand it. Utterance-2, on the other hand, is convoluted because it is phrased as a declarative statement instead of a question. Additionally, Maria needs to establish whether drinking too much water is fatal before working out whether Jemima wants to have lunch. This example is quite artificial. Typically, the messages a sender transmits about a subject or multiple topics are not simultaneous. However, a receiver may have to contend with more than one message to understand a sender's meaning on a particular subject. According to relevance theory, when a receiver has several inputs to appraise, the receiver performs an intuitive comparison of the utterances at hand [12]. The receiver then processes and derives the sender's meaning by drawing on the inputs whose characteristics mostly resemble utterance-1: inputs that require the least effort by the receiver to decipher what the sender means.

Drawing on the principles just described, we theorize that interviewees are likelier to more deeply process information-seeking utterances that they perceive to embody the following characteristics. (a) Clarity: that is, inputs whose substantive meanings are discernible with minimal effort. (b) Worthwhileness: these are utterances an interviewee estimates to contribute an essential difference to deciphering an interviewer's information objectives beyond what the interviewee currently knows. All things being equal, one can reasonably evaluate the worthwhileness of a message if the utterance is sufficiently clear. This premise leads us to offer a proposition similar to Wilson & Sperber's [12]. When multiple information-seeking utterances on a topic are adequately clear, the interviewee will use the messages judged to be the most worthwhile to decipher the interviewer's information objectives. Consequently, interviewees will tend to mentally organize the information they possess into item designations corresponding to information-seeking utterances perceived to be more worthwhile.

## 3.2. Situating the present theory

Existing research works on memory reporting in investigative interviewing are related to the subject matter of the present theory. Others have also examined interviewers' selection of question types when strategizing to elicit information (e.g. [17]). For brevity's sake, we will focus the discussion here on arguably the most influential one to date: the cognitive interview (CI). The CI is an interviewing approach designed to enhance the memory and recall of eyewitnesses (see [18] for an in-depth discussion). CI research addresses disclosure and non-disclosure at the broad or macro level. That is, what makes an interviewee report accurate details on a topic, assuming the interviewee is willing to report on the topic. The effect of questioning strategies is also an aspect at the macro level.

The present theory tackles interviewees' cognition at the micro level: what happens before, for example, mnemonic devices or strategic question types might improve the accuracy of disclosure. The present theory explains how interviewees hear from a question the thing that is being questioned (see also [10]). Hearing what is being questioned from a question is a pragmatic matter; here, the interviewee's problem is not yet an issue of memory and recall. Our contention is that hearing what a question is questioning—pragmatic considerations—is the precursor which mentally enters information items into play[4] in an interview. Then assuming an interviewee has decided to be fully cooperative and disclose everything in question, memory enhancers may help the interviewee elaborate. A resistant interviewee may refrain from disclosing the item(s) a question enters into play. And a semi-cooperative interviewee may or may not disclose those items depending on further strategic considerations. For instance, a semi-cooperative interviewee could think: the thing being questioned is too risky to disclose or OK to reveal. The current theory proposes that the cooperative,

---

[4]We use the phrase enter into play to mean the interviewee mentally flagging an item designation, which they may or may not disclose.

the semi-cooperative and the resistant interviewee first contend the same problem: *what is the thing being questioned?* Pragmatic considerations then mentally enter information item designations into play. Those mental designations may then be affected by mnemonic devices like the CI or strategic considerations to determine disclosure or non-disclosure.

An illustration to further clarify the distinctions just described. Recall again the earlier example about the informant who has just discovered a criminal gang under investigation *smuggles oxycodone into a prison using ambulances*. The interviewee could truly think any of the following item designations might be informative.

(z1) They smuggle oxycodone into a prison using ambulances.
(z2) They smuggle oxycodone into a prison.
(z3) They smuggle oxycodone.
(z4) They smuggle oxycodone using ambulances.

The interviewer interested in knowing about *where* the gang currently peddles narcotics could truly ask question-*i* or -*ii*.

 i. Have you discovered anything about the gang's narcotics operations?
 ii. Have you discovered anything about where the gang peddles narcotics?

Arguably, question-*ii* versus -*i* is a higher-worthwhileness question in terms of the interviewer's objective to know *where* the gang peddles narcotics. Question-*ii* better specifies the interviewer's objective. What the present theory proposes is that a question form like question-*ii* will enter the most pragmatic mental designation into play; here, z2—they smuggle oxycodone into a prison. Then assuming the informant wants to disclose what they have mentally flagged, a memory-enhancing device like the CI might help the informant elaborate.

Note that a question form like question-*i* is not necessarily better or worse than question-*ii*, as a strategic or tactical interviewing matter. An interviewer might deliberately employ question-*i* supposing the interviewer wanted to conceal the objective to specifically discover where the gang peddles narcotics. Nonetheless, posing a question form like question-*i*, which hardly specifies a clear objective, will enter a varied selection of item designations into play between different interviewees. One interviewee could truly think the interviewer wants to know z1. Another interviewee could also truly think the interviewer wants to know z3, and so forth. Assume an interviewee mentally selects z3 as their preferred pragmatic designation. Then a memory-enhancing device like the CI might help the interviewee elaborate. The point of note is this: pragmatic considerations first occur at the micro level. Memory and strategic considerations then exert their influence at the macro level. Thus, an interviewer could ask question-*i*, a low-worthwhileness question form, or a high-worthwhileness question form, question-*ii*, when implementing an interviewing approach like the CI.

The postulations of the present theory are well established in pragmatics [9,10,19,20]. However, because we have adapted those general principles to the niched context of intelligence interviewing, it is useful to subject our proposal to an empirical test.

# 4. Empirically examining information item designation: proof of concept

We conducted two identical experiments to examine our theory that question type influences mental designations and disposition affects utterance choices. As such, Study 1 focused on mental designations and Study 2 on utterance choices.

In both studies, participants assumed the role of an intelligence source with one of two dispositions, intending to be *cooperative* or *resistant* when engaging with their interviewer. The interviewer asked 10 questions on various topics: the interviewees' discoveries about a criminal gang under investigation. Half of the questions specified an objective (*high-worthwhileness condition*), the other half did not specify a clear objective (*low-worthwhileness condition*).

How participants engaged with the interviewer's questions depended on the respective studies. In Study 1, *the designation experiment*, participants did not provide direct answers to the questions; instead, they were instructed to indicate what they think the interviewer wants to know—about their discovery. In doing so, participants flagged their preferred mental designation of an information item. Then participants rated their confidence that their preferred item designation is indeed what the

interviewer wants to know. Conversely, participants in Study 2, *the utterance experiment*, provided direct answers to the interviewer's questions. Participants indicated what they want to say in response to the interviewer's questions.

Suppose the theory under contention has verisimilitude: then the subsequent predictions should receive support, notably the core predictions. Study 1 examined Core Hypotheses 1a, 1b and the Auxiliary Hypothesis. Study 2 examined Core Hypotheses 2a and 2b.

The core of the present theory is that question worthwhileness predicts mental information item designation, regardless of disposition. Thus—in Study 1—the high-worthwhileness questions should elicit a greater preference for item designations that pragmatically correspond to the interviewer's specified objective (Core Hypothesis-1a). That difference in preference should manifest irrespective of whether the participant is a cooperative or resistant source (Core Hypothesis-1b).

One auxiliary hypothesis of the present theory is that high- versus low-worthwhileness questions allow interviewees to better identify the interviewer's objective. That assumption implies that high-worthwhileness questions will allow the interviewee to be more certain about what the interviewer wants to know. Hence, the high- versus low-worthwhileness condition should elicit greater confidence that the corresponding preferred item designations are indeed the things the interviewer wants to know. That difference in preference should manifest irrespective of whether the participant is a cooperative or resistant source (Auxiliary Hypothesis).

In Study 2, which examined utterance, the disposition variable should determine participants' choices. Cooperative versus resistant sources should offer answers that pragmatically correspond to the interviewer's specified objective (Core Hypothesis-2a). Cooperative sources should want to assist the interviewer by disclosing the information the interviewer wants to know. Cooperative sources should be susceptible to the influence of high- versus low-worthwhileness questions; but resistant sources should not be susceptible (Core Hypothesis-2b). Resistant sources should refrain from providing pragmatic disclosures regardless of whether the interviewer asks high- or low-worthwhileness questions. Such sources should want to refrain from assisting the interviewer irrespective of the extent to which the interviewer's question specifies an objective. Terrorist suspects who decline to cooperate abstain from disclosure using behaviours like refusing to comment or claiming to have a lack of memory [7].

## 4.1. Method

### 4.1.1. Participants and design

We aimed to remain with a maximum of $N = 400$, 200 per study, participants (age $\geq 18$ years, balanced across sex) after excluding the data of participants who failed the decision-making instructional manipulation check (IMC) and those who failed one control question, respectively. Thus, we programmed the sampling process in Prolific® to continue sampling until we had 400 participants who had passed all the attention checks and the IMC. That sampling process, which was beyond our control after launching the project, captured 409 participants. Study 1, the designation experiment, included 210 participants, and Study 2, the utterance experiment, comprised 199 participants. The experiments employed the same design: 2 (Question-type: low- versus high-worthwhileness; within-subjects) × 2 (Disposition: cooperative versus resistant; between-subjects).

We recruited English speaking participants with an approval rating of above 90% (compensation = £8/h). The age of the respondents ranged from 19 to 70 years ($M = 27$, $Mdn = 25$). One hundred and three participants stated their preferred pronoun as she/her, 102 chose they/them, 200 he/him, four chose not to disclose their preferred pronoun. Apart from English, participants spoke different languages, majority speaking languages local to Europe or South America (e.g. Polish, Greek, Italian, Portuguese, Spanish). The minority of participants specified languages spoken in countries such as South Africa, Zimbabwe and Bangladesh (e.g. Sesotho, Xhosa, Shona, Bengali). One hundred and forty-eight participants chose not to disclose their native language.

Resource constraints determined our maximum sample size of 200 participants per experiment. But we conducted simulations to examine the level of precision the chosen sample size can provide, given our planned hypotheses tests. The simulations indicated that 200 participants per experiment sufficed to reach the precision we desire under two conditions: (i) a scenario including a wide range of estimated effect sizes using a random draw of response probabilities and (ii) a scenario where there is no effect. The analysis plan provides a detailed description.

### 4.1.2. Procedure

Both experiments employed a similar procedure and materials. Hence, it is necessary to preempt potential treatment diffusion regarding designation and utterance choices. We conducted Studies 1 and 2 simultaneously, using the same Web link to randomly assign prospective participants to the respective studies. This approach ensured that prospective participants partook in either Study 1 or 2, not both. The experiments were entirely online, and we introduced them as studies about communication within a law enforcement context. A Regional Ethics Review Board in Sweden has indicated that this research design does not require a full ethics review (DNR: 812-12). The research adheres to the guidelines of the Swedish Research Council. Before commencing the research, participants provided informed consent to the procedure and received a full debriefing upon completion. The appendix contains the full details of all the materials described in the procedure.

For the sake of conciseness, what follows describes the procedure for Studies 1 and 2. When necessary, we highlight the differences in protocol between the studies

**Phase 1: intelligence-source role.** By reading a short background story, participants assumed the role of a source who can gather information about a criminal gang. The story mimicked the typical intelligence scenario whereby sources contend the *possibility* of disclosing information to an interviewer. Research studies usually employ background stories to create such source roles (e.g. [8]). The story manipulated sources' dispositions in two ways. Half of the participants assumed a mindset to be a cooperative source when engaging with their interviewer. The other half assumed the mindset of a resistant source when engaging with their interviewer. We included a manipulation check to assess the efficacy of the disposition manipulation (see appendix A).

We did not include a semi-cooperative condition for analytic reasons. Three disposition conditions will bring overly complex models to our hypothesis tests and obscure interpretation. Using the cooperative and resistant conditions is prudent because a semi-cooperative disposition includes both cooperative and resistant mindsets. Thus, the present study still contributes to understanding how semi-cooperative sources might behave when acting on either a cooperative or resistant disposition. Additionally, the present studies are initial ones in this line of research. For that reason, we believe it is prudent to employ a parsimonious design at this time; this issue is revisited in the analysis plan.

**Phase 2: Decision-making instructions.** Next, participants received instructions on how to engage with the interviewer's questions. The instructions depended on the study participants are randomly assigned to undergo. We included an IMC to identify and exclude inattentive participants who fail the check (see appendix B).

*Study 1: Mental designation.* In this experiment, participants received instructions about indicating their preferred mental designation of an information item. Owing to the nature of the theoretical propositions under examination, it was necessary to ensure that participants indicate their preferred pragmatic designation—and not the information they necessarily intend to disclose. Thus, the instructions told participants that upon receiving a question from their interviewer, they (i.e. participants) were to indicate what they thought the interviewer *wanted to know*—not necessarily what they intend to disclose.

*Study 2: Utterance.* In this experiment, the instructions told participants to provide direct answers to the interviewer's questions. Thus, participants indicated what they wanted to say in response to each question.

**Phase 3: Discoveries, questions and decisions.** Here, participants underwent ten scenarios presented in random order. Appendix C contains all the material described here. In each scenario, participants first made a discovery about the criminal gang under investigation. Each discovery contained three parts that describe the details therein in ascending order: (a) bare minimum details; (b) medium details, comprising a new detail plus the bare minimum; and (c) complete details, consisting of a new detail including the bare minimum and medium details. For example, *you have noticed that the drug deals happen in the evening* (bare minimum) *when the workday ends* (medium) *at 18.00* (complete). Thus, each discovery produced three legitimate information items, also describing the discovery's details in ascending order. For example,

*the drug deals happen in the evening* (bare minimum);
*the drug deals happen in the evening when the workday ends* (medium);
*the drug deals happen in the evening when the workday ends at 18.00* (complete).

After each discovery, the interviewer posed either a high- or low-worthwhileness question, five questions per condition. In the high-worthwhileness condition, the questions specified an objective. Those questions asked for a *specific thing*—the complete details about the discovery. Each question in

the low-worthwhileness condition was ambiguous: such questions asked for *anything* about the discovery under investigation. Thus, the low-worthwhileness question could reasonably elicit the bare minimum, medium or complete details.

How we invited participants to engage with each question depended on the study to which participants are assigned.

**Study 1: Mental designation.** In this experiment, participants were instructed to flag their preferred mental designation of an information item—what they thought the interviewer wanted to know. They indicated that preference by choosing any of the three options describing their discovery's bare minimum, medium, or complete details. Participants were also provided the option to indicate that they cannot determine what the interviewer wants to know.

Participants who selected an information item went on to provide two ratings, in random order, examining their confidence in their choice. A mandatory rating directly asked participants how confident they were that their preference is indeed what the interviewer wanted to know (1 = *not confident at all*, 5 = *completely confident*). An optional rating asked participants to stake a hypothetical bet that their preference is what the interviewer wanted to know; the wager was a percentage of their compensation (0% = *none of my compensation*, 100% = *all of my compensation*).

**Study 2: Utterance.** In this experiment, participants were instructed choose what they wanted to say in response to the interviewer's question. They decided by choosing any of the three options describing their discovery's bare minimum, medium or complete details. We included the option to respond by choosing 'no comment': suppose a participant wished to remain silent regarding the interviewer's question. Unlike Study 1, participants who selected an information item did not provide confidence ratings since confidence in utterance was not germane to our theory. Moreover, implementing the confidence assessment in Study 2 was likely to confuse participants about their main task.

In both experiments, we included four control questions to flag the data of inattentive participants (see appendix D).

# 5. Analysis plan

We examined the *core hypotheses* using Bayesian categorical regression models. The analysis produced posterior distributions over parameters quantifying the probability of each possible parameter value given the data. We report the posterior mean with the corresponding 95% credible interval (95%-CrI) and the 95% highest density interval (HDI). The 95%-CrI is the range around the posterior mean within which the true value of the parameter lies with a probability of 0.95. The HDI is identical to the CrI if the posterior is symmetric; if the posterior is asymmetric, the endpoints of both intervals may differ.

Following [21,22], we defined a region of practical equivalence (ROPE) to test whether we found evidence consistent with our predictions. The ROPE can be understood as a null region. We reject the null hypothesis if the parameter's HDI falls outside of the null region. If the parameter's HDI overlaps with the null region and the sign is positive, we reject a theory postulating a negative effect. If the sign is negative, we reject a theory postulating a positive effect. If the parameter's HDI falls within the null region, we conclude that the data are consistent with 'no effect' (not to say that we have proved that the null hypothesis is true). We do settle on a conclusion from our data when the ROPE lies entirely within the parameter's HDI. Furthermore, the ROPE served as a stopping rule for testing: once decided on the null region, we collected data until

(a) the 95% HDIs of the parameters of interest are, at most, as wide as the null region, or
(b) we reached our maximum participant sample size of 400 (200 per experiment) due to resource constraints.

Given our planned hypotheses tests, we conducted simulations to examine the level of precision the chosen sample size can provide. Our desired level of precision was that the width of the coefficients' 95% HDIs should be equal to or smaller than 0.5. The utterance experiment (Study 2) served as the benchmark because that study includes four relevant outcome levels: no comment, bare details, medium details and complete details. The designation experiment (Study 1), on the other hand, involves three relevant outcome levels: bare details, medium details, and complete details. Hence, Study 2 versus Study 1 might require a larger sample size to achieve acceptable precision even though both studies employ a similar design.

We ran four models with simulated data (see electronic supplementary materials: Rmarkdown documents DataSimulation 1 and DataSimulation 2). The simulations indicated that our hypotheses tests require a sample size greater than 150 participants per experiment. A model with 200 participants reached our desired precision for two different scenarios. DataSimulation 1: a wide range of estimated effect sizes using a random draw of response probabilities; DataSimulation 2: a scenario where there is no effect. One can access the simulations in exhaustive detail here: https://osf.io/7tgwe/.

## 5.1. Model specification

### 5.1.1. Fixed and random effects

To predict designation (Model 1) and utterance (Model 2) preferences, we fit two Bayesian categorical regression models using the R [23] package brms [24]. The package provides an interface to fit Bayesian mixed models using Stan [25].

For Model 1, the variables disposition (cooperative versus resistant) and question type (high- versus low-worthwhileness) were included as predictors. We run a model including an interaction of both predictors for exploratory purposes. The dependent variable was the probability of choosing *designations* with medium details, complete details, or the option not to specify a designation over designations with bare details. Model 2 included disposition as a main effect. And we added question type as a nested effect to examine the effect of question type at the two levels of disposition (cooperative and resistant sources). The dependent variable was the probability of choosing *utterances* with medium details, complete details, or no comment over utterances with bare details. For both Models 1 and 2, designations and utterances with bare information was the reference category. Both models included varying intercepts and slopes for participants and scenario items. We implemented that inclusion assuming (1) the effect of question type on designation and utterance choices varies between participant and scenario item, and (2) the effect of disposition on designation and utterance choices varies between scenario item.

### 5.1.2. Priors

For both models, we used the same weakly regularizing priors, which allow a reasonably wide range of parameter values and penalize very extreme values. The priors for the by-designation and -utterance intercepts were normal distributions with mean 0 and standard deviation 3. For fixed effects, normal priors with a mean of 0 and a standard deviation of 1 were used. Random effects were modelled as a correlation matrix and a vector of standard deviations. The standard deviations were assigned half-normal priors with a mean of 0 and a standard deviation of 1. For the correlation matrix, an LKJ(2) prior was used such that smaller correlations are favoured over extreme values such as ±1 [25,26]. A prior-sensitivity analysis was carried out to assess whether priors are dominating the posterior distribution. We specifically contrasted the aforementioned models with models having the following less conservative prior specifications.

Intercept N(0, 10)/student_t(3, 0, 2.5)
Fixed effects N(0, 1)/flat prior
Random effects sd Half Normal(0, 1)/student_t(3, 0, 2.5)
Correlation LKJ(2)/LKJ(1)

## 5.2. Predictions

### 5.2.1. Model 1: designation

We expected Core Hypothesis-1a to produce the following result. High-worthwhileness questions should elicit a preference for participants to choose designations with complete and medium details over designations with bare details, and designations with bare details over the option to not choose any designation. There should be no effect of disposition on the participants' item designation preferences (Core Hypothesis-1b).

### 5.2.2. Model 2: utterance

We expected Core Hypothesis-2a to produce the following result. Cooperative versus resistant sources should more frequently choose utterances with complete and medium details as opposed to

utterances with bare details or the no-comment option. According to Core Hypothesis-2b, cooperative sources should be prone to the influence of high- versus low-worthwhileness questions; but resistant sources should be uninfluenced by question type (Core Hypothesis-2b).

### 5.2.3. Summary of predictions

In all, the question-type versus disposition manipulation should influence participants' choices to a greater extent in Study 1 (the designation experiment). Conversely, the disposition versus question-type manipulation should influence participants' choices to a greater extent in Study 2 (the utterance experiment).

## 5.3. Region of practical equivalence and model comparison

Owing to the novelty of our study design, particularly the intelligence interviewing aspect, we were unable to find directly related studies. To our knowledge, no research has examined how question-worthwhileness and disposition differentially influence mental designation and utterance, much less with categorical regression. Some relevant studies include Lorson *et al.* [27,28] that examined the effect of disposition (i.e. cooperation) on utterance choices. That research found cooperation effects ranging from 0.54 to 1.68 in log-odds. Hence, for the predictor *disposition*, we assumed the following range of log-odds as ROPE: −0.25 to 0.25 (width of 0.5). We specify the same ROPE [−0.25 to 0.25] for the predictor *question type* and the *interaction terms*.

Note that our specified ROPE range was more conservative than the default ROPE range of [−0.18, 0.18] based on [21] and negligible effect sizes for behavioural sciences according to Cohen [29]. We hoped that the present study would assist future studies in specifying effect sizes and ROPE specifications. Suppose we were to follow the recommended heuristic in [30]. In that case, we would have chosen the lower limit of the 95% CI of effects found in similar studies, for example, Lorson *et al.* [27]. Then a negligible effect would be anything less extreme than 0.1. However, given our resource constraints, we would be unable to reach a precision with HDIs equal to or smaller than 0.2 in range.

Suppose we decided to use the bottom limit of the 95% CI for the combined disposition effect from the studies mentioned earlier instead of examining the smallest possible effect (i.e. 0.54). In that case, based on a combined 95% CI of [0.24, 0.83], we would yield a very similar ROPE to the one specified above ([−0.24, 0.24] instead of [−0.25, 0.25]). That observation led us to believe our specified ROPE [−0.25, 0.25] is prudent at this time. That notwithstanding, we acknowledge that we run the risk of missing plausible effects that are smaller than the limits of our ROPE.

### 5.3.1. Sampling process

Samples would be drawn from the posterior distributions of the model parameters using the NUTS sampler [31]. We will run four sampling chains, each collecting 4000 iterations, whereby the first 1000 iterations will be disregarded as part of the warm-up phase leading to 12 000 iterations available for analysis. This sampling process was the same for all models and the chains mixed well (all $R = 1.0$).

## 5.4. Secondary analysis: auxiliary hypothesis

The auxiliary hypothesis was also examined within the Bayesian framework. We report the posterior mean and the 95% credible interval (95%-CrI) and the probability that a given coefficient is greater than zero given the data and model. According to the auxiliary hypothesis, the high- versus low-worthwhileness condition should elicit greater confidence that corresponding preferred item designations are indeed the things the interviewer wants to know. Importantly, that difference in preference should manifest irrespective of disposition. Next follows the model specifications.

### 5.4.1. Fixed and random effects

We examined participants' confidence ratings by fitting a mixed-effects Bayesian ordinal (cumulative) regression model (Model 3). Question type and disposition were included to predict confidence. The model included varying intercepts and slopes for participants and scenario items, assuming that the effect of question type on confidence ratings varies between participants and scenarios.

### 5.4.2. Priors

We used weakly regularizing priors, allowing a reasonably wide range of parameter values and at the same time penalized very extreme values. The priors for the intercept were normal distributions with mean 0 and standard deviation 3. For both fixed effects, we used normal priors with a mean of 0 and a standard deviation of 1. Random effects were modelled as a correlation matrix and a vector of standard deviations. The standard deviations were assigned half-normal priors with a mean of 0, and a standard deviation of 1. For the correlation matrix, an LKJ(2) prior was used such that smaller correlations are favoured over extreme values such as ±1. A prior-sensitivity analysis was carried out to assess whether priors are dominating the posterior distribution. We contrasted the aforementioned model with a model having the following less conservative prior specifications (see appendix III):

Intercept student_t(3, 1.6, 2.5)
Fixed effects flat prior
Random effects sd student_t(3, 0, 2.5)
Correlation LKJ(1)

### 5.4.3. Second confidence measure: willingness to bet

We included a second confidence measure to examine the auxiliary hypothesis: the willingness of participants to place a bet that their preference is what the interviewer wanted to know. Similar to the confidence ratings, high- versus low-worthwhileness questions were predicted to increase the probability of betting, independent of disposition. We examined the prediction using a mixed-effects Bayesian logistic regression model (Model 4).

**Fixed and random effects.** Question type and disposition were included to predict the probability of betting. The model included varying intercepts and slopes for participants and scenario items, assuming that the effect of question type on the participants' confidence varies between participant and scenario items.

**Priors.** Again, we used weakly regularizing priors, allowing a reasonably wide range of parameter values and penalizing very extreme values. The prior for the intercept is a normal distribution with mean 0 and standard deviation 3. For fixed effects, normal priors with a mean of 0 and a standard deviation of 1 were used. Random effects were modelled as a correlation matrix and a vector of standard deviations. The standard deviations were assigned half-normal priors with a mean of 0, and a standard deviation of 1. For the correlation matrix, an LKJ(2) prior was used such that smaller correlations are favoured over extreme values such as ±1. A prior-sensitivity analysis was implemented to assess whether priors are dominating the posterior distribution. We specifically contrasted the aforementioned model with a model having the following less conservative prior specifications:

Intercept N(0, 10)/student_t(3, 0, 2.5)
Fixed effects N(0, 1)/flat prior
Random effects sd Half Normal(0, 1)/student_t(3, 0, 2.5)
Correlation LKJ(2)/LKJ(1)

## 6. Study design template

**Notes:**

1. We coded the outcome measure (information items) as follows (with 'bare' as reference level); the coding was the same for all the relevant analyses:

|  | complete | medium | no comment |
| --- | --- | --- | --- |
| bare | 0 | 0 | 0 |
| complete | 1 | 0 | 0 |
| medium | 0 | 1 | 0 |
| no comment | 0 | 0 | 1 |

1. **Sampling plan and test sensitivity rationale:** We aimed to include a minimum of $N = 400$ participants, $N = 200$ per study. Resource constraints and the lack of previous research (to precisely estimate an ROPE) determined our sample size choice. For Studies 1 and 2, it holds that we cannot conclude anything from our data when the ROPE lies entirely within the parameter's HDI.

2. **Theory that could be shown wrong by outcomes:** This research is a first test of the theory outlined in the article. Thus, we cannot disprove the theory, at least not with certainty at this time. And we cannot rule out plausible effects that are smaller than the limits of our ROPE. Nonetheless, supporting or rejecting the respective core hypotheses will count for and against the theory's corresponding postulations. This approach assists in potentially refining the theory and developing replication studies.

**Table X1**

Study 1 Designation Experiment.

| hypothesis | Model I | analysis | predictions |
|---|---|---|---|
| **Core Hypothesis-1a** high-worthwhileness questions should elicit a greater preference for item designations that pragmatically correspond to the interviewer's specified objective | brm(Response ~ Disposition + QuType + (QuType \| SubjectID) + (Disposition + QuType \| Context)) <br><br> contrast coding for Model I: <br> Question Type: <br> high-worthwhileness: 1 <br> low-worthwhileness: −1 <br><br> Disposition: <br> cooperative: 1 <br> resistant: −1 <br><br> the model for Study 1 included the two predictors Question Type and Disposition and no interaction term. A model including an interaction term was run for exploratory purposes | to test this hypothesis, we investigated whether there was a main effect of Question Type on the log-odds of choosing complete/ medium/no comment designations over bare information designations | P(designation = complete/medium) > P(designation = bare) <br><br> the Question Type parameter's HDI—for both P(complete > bare) and P(medium > bare)— should lie outside the ROPE and have a positive sign for high-worthwhileness questions (which are coded as 1) <br><br> P(designation = No comment) > P(designation = bare) <br><br> the Question Type parameter's HDI should lie outside the ROPE and should have a negative sign for high-worthwhileness questions (which are coded as 1) |

(*Continued.*)

| hypothesis | Model I | analysis | predictions |
|---|---|---|---|
| **Core Hypothesis-1b**<br>there should be no effect of Disposition on the participants' item designation preferences | | to test this hypothesis, we investigated whether there was a main effect of Disposition on the log-odds of choosing high/medium/no comment designations over bare information designations | all the Disposition parameter's HDIs are predicted to fall within the null region, such that we conclude that the data are consistent with 'no effect' of Disposition (not to say that we have proven that the null hypothesis is true) |

**Table X2**

Study 2 Utterance Experiment.

| hypothesis | Model II | analysis | predictions |
|---|---|---|---|
| **Core Hypothesis-2a** cooperative sources should more frequently choose utterances with complete/medium informativeness as opposed to bare informativeness or the no-comment option than resistant sources | brm(Response ~ Disposition/QuType + (QuType \| SubjectID) + (Disposition/ QuType \| Context) contrast coding for Model II: Question Type: high-worthwhileness: 0.5 low-worthwhileness: −0.5 | to test this hypothesis, we investigated whether there was a main effect of Disposition on the log-odds of choosing complete/medium/no information utterances over bare information utterances, holding the predictor Question Type constant | P(utterance = complete/medium) > P(utterance = bare) the Disposition parameter's HDI—for both P(complete > bare) and P(medium > bare)— should lie outside the ROPE and should have a positive sign for cooperative sources |
| **Core Hypothesis-2b** only cooperative sources as opposed to resistant sources should be affected by the predictor Question Type → *cooperative sources should be influenced by the predictor Question Type* → *resistant sources should not be influenced by the predictor Question Type* | Disposition: cooperative: 0.5 resistant: −0.5 parameter estimated: complete_Intercept complete_Disposition1 complete_Dispositioncooperative: QuType1 complete_Dispositionresistant: QuType1 | to test Hypothesis-2b we investigated whether there was a simple effect of Question Type on the log-odds of choosing high/medium/no detail utterances over bare detail utterances for cooperative versus resistant sources | P(utterance = no comment) > P(utterance = bare) the Disposition parameter's HDI should lie outside the ROPE and should have a negative sign for cooperative sources Cooperative Group (Dispositioncooperative:QuType): |

**Table X2** (*Continued.*)

| hypothesis | analysis | predictions |
|---|---|---|
| | **Model II** | |
| | medium_Intercept | $P(\text{utterance} = \text{complete/medium/no comment}) > P(\text{utterance} = \text{bare})$ |
| | medium_Disposition1 | |
| | medium_Dispositioncooperative: QuType1 | the Question Type parameter's HDI should lie |
| | medium_Dispositionresistant: QuType1 | outside the ROPE such that there is a difference |
| | | between seeing a high- versus low- |
| | nocomment_Intercept | worthwhileness question for cooperative sources |
| | nocomment_Disposition1 | |
| | nocomment_Dispositioncooperative: QuType1 | |
| | nocomment_Dispositionresistant: QuType1 | Resistant Group (Dispositionresistant: QuType1) |
| | | |
| | this model tests the main effect of Disposition | all the Question Type parameter's HDIs are |
| | (Disposition1) averaged over the levels of Question Type, | predicted to fall within the null region, such |
| | and the effect of Question Type within the group of | that we conclude that the data are consistent |
| | cooperative sources (Dispositioncooperative:QuType1) and | with 'no effect' of Question Type for resistant |
| | resistant sources (Dispositionresistant:QuType1). Thus, the | sources (not to say that we have proven that |
| | model tests whether the two levels of Question Type differ | the null hypothesis is true) |
| | for cooperative versus resistant sources | |

## 6.1. Auxiliary hypothesis: confidence in mental designations (Study 1)

**Table X3**

*Confidence ratings.* Output (5-point Likert scale): 'not confident at all', 'slightly confident', 'somewhat confident', 'fairly confident', 'completely confident'.

| hypothesis | Model III (Ordinal cumulative model) | analysis | predictions |
|---|---|---|---|
| the high- versus low-worthwhileness condition should elicit greater confidence, but Disposition should not influence the ratings | brm(Likert ~ QuType + Disposition + (QuType \| SubjectID) + (QuType + Disposition \| Context)) <br><br> contrast coding for Model I: <br><br> Question Type: <br>    high-worthwhileness: 1 <br>    low-worthwhileness: −1 <br><br> Disposition: <br>    cooperative: 1 <br>    resistant: −1 | to test this hypothesis, we investigated whether there was a main effect of Question Type on the confidence ratings | the Question Type parameter's HDI should lie outside the ROPE and should have a positive sign for high-worthwhileness questions (which are coded as 1). <br><br> the Disposition parameter's HDI is predicted to fall within the null region, such that we conclude that the data are consistent with 'no effect' of Disposition on the confidence ratings (not to say that we have proven that the null hypothesis is true) |

**Table X4**

*Willingness to bet.* Output: willingness to bet, 'yes' (1), 'no' (0).

| hypothesis | Model IV (Ordinal Cumulative model) | analysis | predictions |
|---|---|---|---|
| high- versus low-worthwhileness questions should increase the probability of betting, independent of disposition | brm(Bet ~ QuType + Disposition + (QuType \| SubjectID) + (QuType + Disposition \| Context) | to test this hypothesis, we investigated whether there was a main effect of Question Type on the participants' willingness to bet | the Question Type parameter's HDI should lie outside the ROPE and should have a positive sign for high-worthwhileness questions (which are coded as 1) |
| | | | the Disposition parameter's HDI is predicted to fall within the null region, such that we conclude that the data are consistent with 'no effect' of Disposition on the confidence ratings (not to say that we have proven that the null hypothesis is true) |

# 7. Results

One can access the data supporting the results, full analysis code, and supplemental material here: https://osf.io/bgxrj/.

## 7.1. Study 1: mental designation

As depicted in figure 2, participants in the cooperative and resistant conditions preferred complete details. But that preference was stronger for high- than low-worthwhileness questions.

We reached our desired level of precision (i.e. the width of the coefficients' 95% HDIs should be equal to or smaller than 0.5) for two coefficients: the question-type slope coefficient for complete detail designations as well as the disposition slope coefficient for medium detail designations; see bold text in table 1. Focusing on question type, Model 1 indicated that for designations with complete details ($b = 0.41$, *CrI*: [0.18,0.65]), high- versus low-worthwhileness questions led to an increase in log-odds for choosing complete detail designations over bare detail designations. These results suggest that high-worthwhileness questions were more likely to elicit complete details than low-worthwhileness questions.

However, the 95% HDI of the same coefficient (question type for designations with complete details) was within the ROPE to 5.92% (figure 3). Thus, we cannot confirm that high-worthwhileness questions elicited a greater preference for mental designations that pragmatically correspond to the interviewer's specified objective (Core Hypothesis-1a). Nonetheless, since the HDI overlaps with the null region and has a positive sign, we can reject a theory that posits a negative effect of

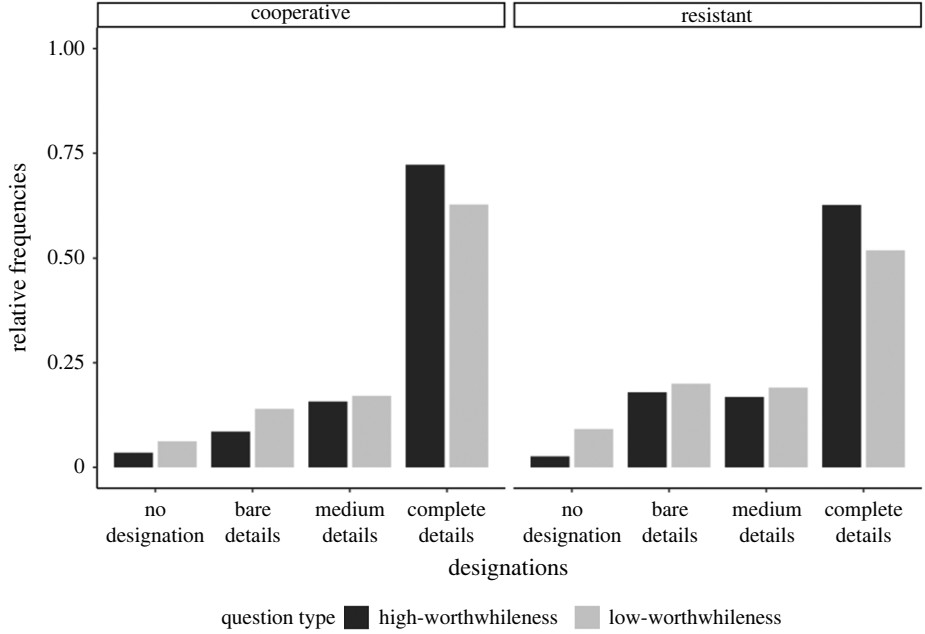

**Figure 2.** Relative frequencies of preferences for the information item designations.

**Table 1.** Population-level estimates of Model 1 in log-odds with the standard errors and 95% credible intervals. By-designation (grand) intercepts are listed first, followed by the slope estimates for Question-type and Disposition. Slope coefficients that reached the desired level of precision are shown in bold. The slope for Question-type is the change in log-odds for the high-worthwhileness question (1, high-worthwhileness; −1, low-worthwhileness) and the slope for Disposition is the change in log-odds for cooperative participants (1, cooperative; −1, resistant).

| detail designations | coefficient | posterior mean | est. error | l-95% Crl | u-95% Crl |
|---|---|---|---|---|---|
| complete | intercept | 2.00 | 0.35 | 1.32 | 2.68 |
| medium | intercept | 0.16 | 0.18 | −0.19 | 0.51 |
| no designation | intercept | −1.99 | 0.36 | −2.74 | −1.31 |
| **complete** | **question type: high- versus low-worthwhileness** | **0.41** | **0.12** | **0.18** | **0.65** |
| medium | question type: high- versus low-worthwhileness | 0.07 | 0.13 | −0.20 | 0.32 |
| no designation | question type: high- versus low-worthwhileness | −0.33 | 0.25 | −0.82 | 0.17 |
| complete | disposition: cooperative versus resistant | 0.50 | 0.22 | 0.07 | 0.93 |
| **medium** | **disposition: cooperative versus resistant** | **0.25** | **0.11** | **0.03** | **0.47** |
| no designation | disposition: cooperative versus resistant | 0.24 | 0.21 | −0.15 | 0.67 |

question type on mental designations with complete details—low-worthwhileness questions do not lead to an increased preference for information with complete details over bare details.

The question-type coefficients' 95% HDI for medium details designations nearly reached precision (width = 0.52) and was largely within the ROPE (to 94.54%), consistent with 'no effect' of

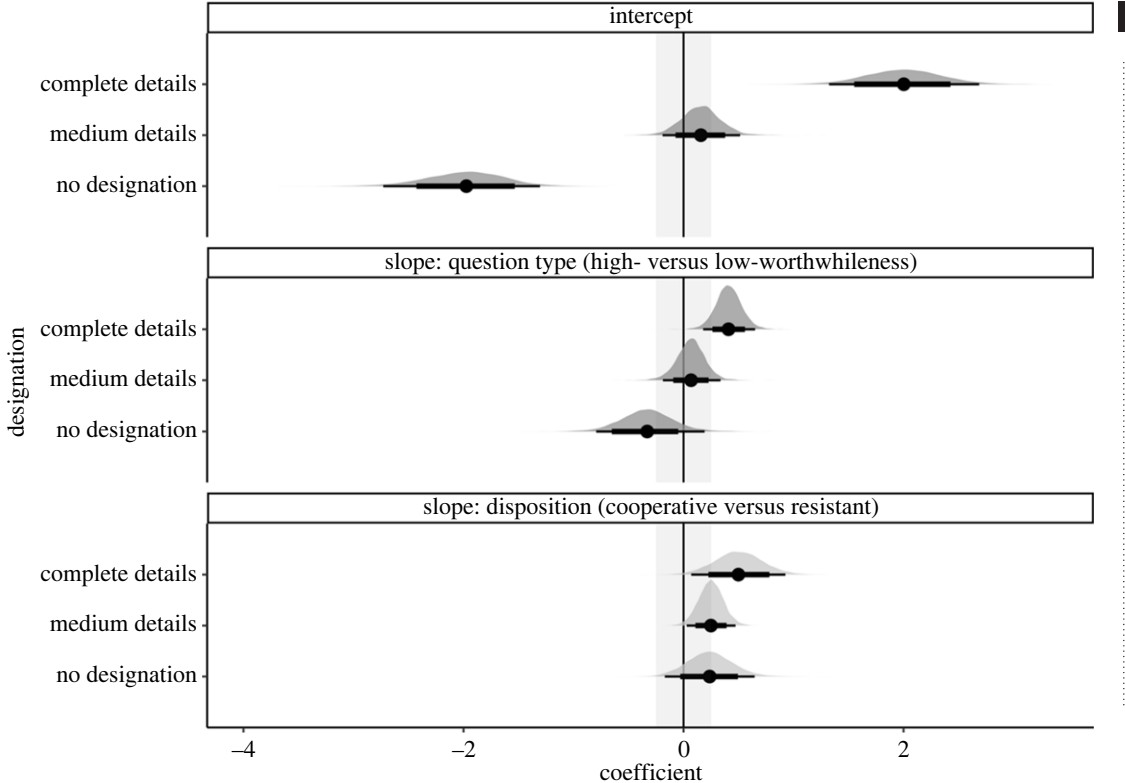

**Figure 3.** Posterior distributions over population-level estimates for Model 1 with 80% and 95% highest density intervals and the ROPE area shaded in light grey.

question type on the preference for designations with medium details. Those results speak against parts of Core Hypothesis-1a that high-worthwhileness questions would lead to an increased preference for mental designations with medium details over bare details. But a replication is needed to confirm that assertion.

Furthermore, Model 1 revealed that for designations with medium details, $b = 0.25$, *CrI*: [0.3,0.47], a cooperative versus resistant disposition led to an increase in log-odds for choosing medium detail designations over bare detail designations. The 95% HDI of that coefficient—the effect of disposition on medium details designations—overlaps with the null region and has a positive sign. Thus, we can reject a theory postulating that resistant versus cooperative interviewees more frequently mentally designated medium details over bare details. That finding suggests that cooperative versus resistant interviewees tend to mentally flag medium over bare details. For the remaining coefficients, we are not able to draw any firm conclusions; the data could not reach precision. Overall, we do not have evidence for and only weak evidence against Core Hypothesis-1b, which posited that disposition does not influence the mental designation of information items.

### 7.1.1. Auxiliary hypotheses

**Confidence in mental designations.** Figure 4 indicates that most participants, across conditions, were completely or fairly confident in the mental designation preferences. Participants were more confident during the high-worthwhileness trials. Resistant interviewees in low-worthwhileness trials were least confident in their mental designation preferences.

The HDIs of both slope coefficients reached the required precisions, coefficients in bold (table 2). As predicted, Model 3 indicates an increase of confidence in mental designations for high- versus low-worthwhileness questions, $b = 0.28$, *CrI*: [0.12, 0.45]. Since the HDI overlaps with the null region and has a positive sign, we can reject a theory that posits a negative effect of question type on the participants' confidence in their designation choice. That result indicates that low-worthwhileness questions do not lead to an increase in the participants' confidence when choosing between designations. In contrast, the HDI for the slope coefficient of disposition fell within the ROPE

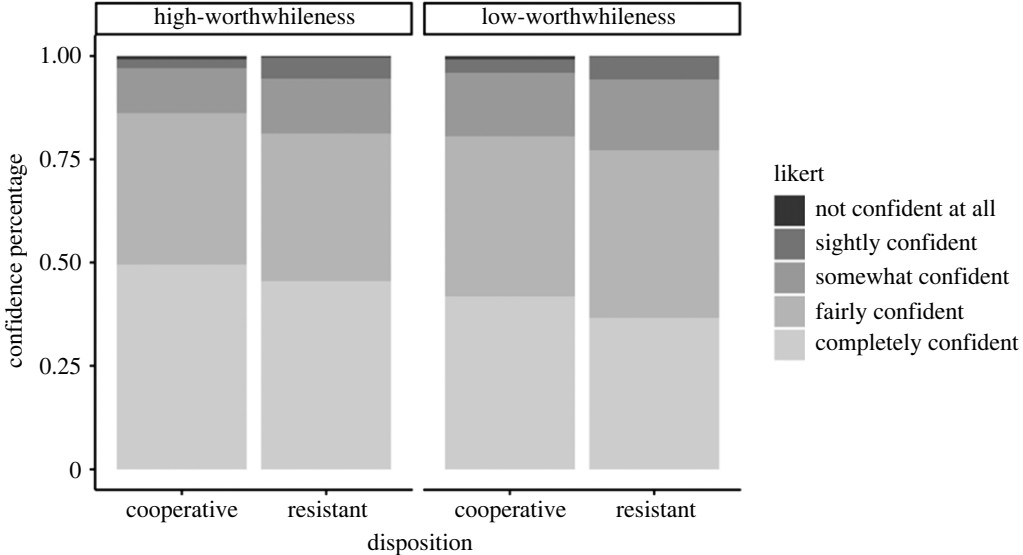

**Figure 4.** Relative percentages of confidence in information item designations.

**Table 2.** Population-level estimates of Model 3 in log-odds with the standard errors and 95% credible intervals. Intercepts are listed first, followed by the slope estimates for Question-type and Disposition. Slope coefficients that reached the desired level of precision are shown in bold. The slope for Question-type is the change in log-odds for the high-worthwhileness question (1, high-worthwhileness; −1, low-worthwhileness) and the slope for Disposition is the change in log-odds for cooperative participants (1, cooperative; −1, resistant).

| estimate | posterior mean | est. error | l-95% CrI | u-95% CrI |
|---|---|---|---|---|
| Intercept1 | −6.68 | 0.39 | −7.46 | −5.94 |
| Intercept2 | −4.31 | 0.26 | −4.83 | −3.79 |
| Intercept3 | −2.30 | 0.24 | −2.76 | −1.83 |
| Intercept4 | 0.48 | 0.23 | 0.04 | 0.95 |
| **question type: high- versus low-worthwhileness** | **0.28** | **0.08** | **0.12** | **0.45** |
| **disposition: cooperative versus resistant** | **0.18** | **0.16** | **−0.14** | **0.50** |

to 66.11% and includes zero which suggests 'no effect' of disposition (cooperative or resistant) on confidence.

**Willingness to bet on mental designations.** Model 4 used a mixed-effects Bayesian logistic regression to examine willingness to wager on preferred mental designations as a function of question type and disposition. The HDIs of both slope coefficients did not reach the desired precisions. Thus, given our data, we cannot make any claims about the effects of question type or disposition on participants' willingness to bet that their preferred mental designations were what the interviewer wanted to know (table 3).

## 7.2. Study 2: utterance

Cooperative interviewees mostly preferred to disclose complete details; the no-comment option was the least preferred. By contrast, resistant interviewees preferred to utter bare details or no comment rather than complete details. Figure 5 illustrates the pattern of preferences but does not indicate a systematic effect of question type on the participants' utterances choices.

The pattern of disclosure preferences aligns with the strategy participants indicated during the disposition check (table 4). Most participants in the cooperative condition indicated they would reveal everything they knew ($n = 60$), and some participants noted they would disclose some but not all the

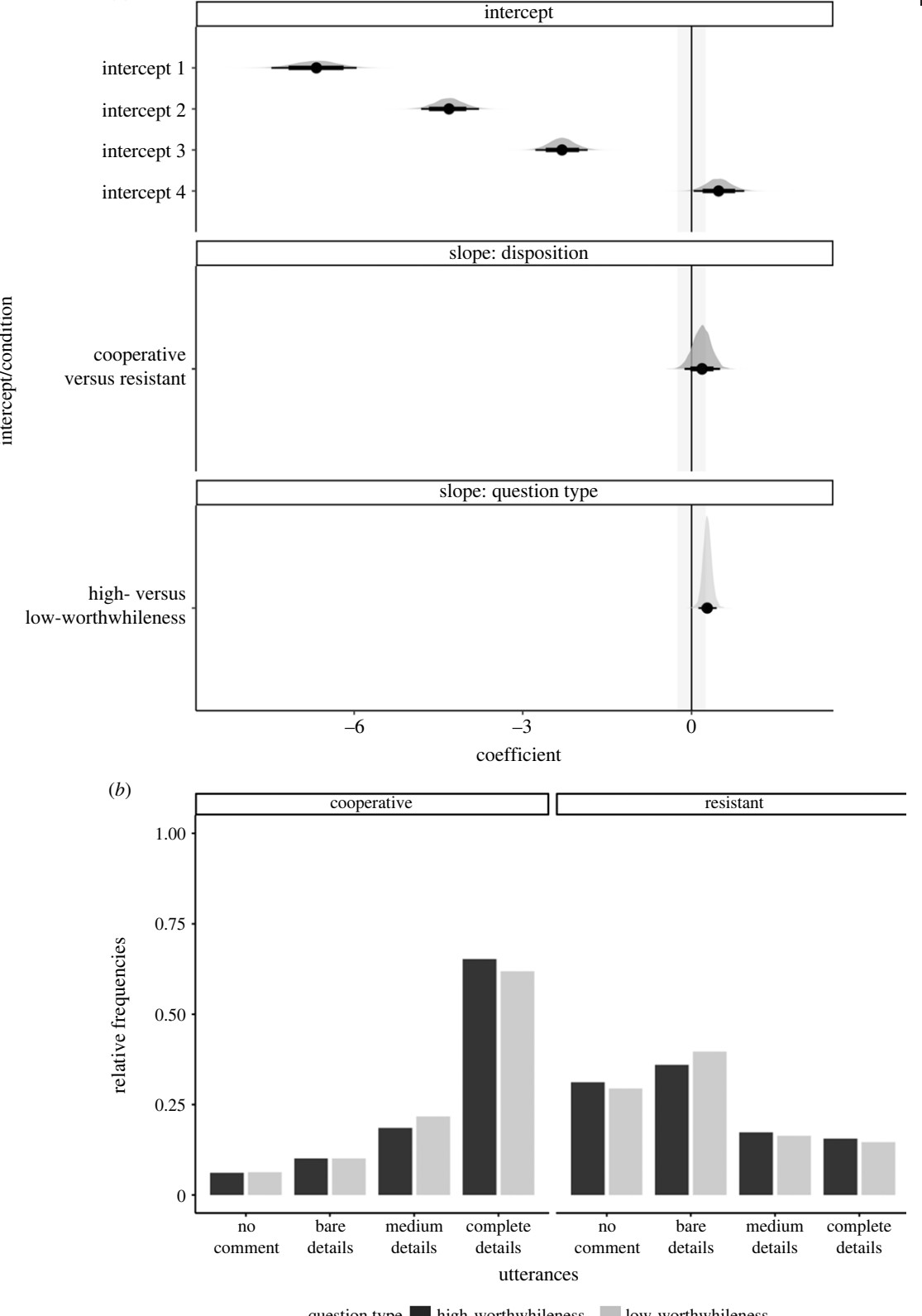

**Figure 5.** (a) Posterior distributions over population-level estimates for Model 3 with 80% and 95% highest density intervals and the ROPE area shaded in light grey. (b) Relative frequencies of preferences for utterances.

information ($n = 31$). Resistant participants indicated they would mostly reveal some but not all information ($n = 55$) or lie ($n = 27$). Since participants were not given the option to lie, it is possible that those who intended to lie opted to stay silent or disclose bare details during the utterance task.

**Table 3.** Population-level estimates of Model 4 in log-odds with the standard errors and 95% credible intervals. Intercepts are listed first, followed by the slope estimates for Question-type and Disposition. Slope coefficients that reached the desired level of precision are shown in bold. The slope for Question-type is the change in log-odds for the high-worthwhileness question (1, high-worthwhileness; −1, low-worthwhileness) and the slope for Disposition is the change in log-odds for cooperative participants (1, cooperative; −1, resistant).

| estimate | posterior mean | est. error | l-95% CrI | u-95% CrI |
| --- | --- | --- | --- | --- |
| intercept | 3.72 | 0.41 | 2.86 | 4.49 |
| question type: high- versus low-worthwhileness | 0.46 | 0.20 | 0.06 | 0.86 |
| disposition: cooperative versus resistant | 0.17 | 0.24 | −0.31 | 0.65 |

**Table 4.** Disposition check ratings grouped by disposition (cooperative versus resistant). Disposition check ratings range from −1 to 2 and show what participants intend to do with the disposition introduction (−1: I will lie, 0: I will keep silent, 1: I will reveal some but not all information, 2: I will reveal everything).

| disposition | mean | s.d. | median | mode |
| --- | --- | --- | --- | --- |
| cooperative | 1.58 | 0.61 | 2 | 2 |
| resistant | 0.48 | 0.99 | 1 | 1 |

None of the regression coefficients' 95% HDIs reached precision (table 5). The nature of this finding prevents us from speaking definitively on Core hypotheses 2a and 2b. However, the 95% HDIs for the effect of disposition on utterance choices with complete versus bare details as well as medium versus bare details were well outside of the ROPE. That result, which sheds light on Core Hypothesis-2b, warrants further examination (figure 6 and italic coefficients in table 5). But we must emphasize that those findings need to be replicated before drawing firm conclusions.

Averaged over question type, Model 2 indicated that cooperative as opposed to resistant interviewees more frequently chose to utter complete details over bare details ($b = 4.26$, *CrI*: [3.29, 5.22], in log odds); see table 5 for more details. Similarly, averaged over question type, cooperative as opposed to resistant interviewees more frequently chose to utter medium over bare details ($b = 1.66$, *CrI*: [1.07, 2.23], in log-odds).

# 8. Discussion

We proposed mechanisms by which intelligence interviewees mentally identify relevant information in an interview. Our contention was that interviewees determine relevant information items based on the extent to which a question specifies an interviewer's information objective—what the interviewer wants to know. The primary hypothesis was the following. High-worthwhileness questions that specify information objectives better predict interviewees' mental designations (of information items) than ambiguous low-worthwhileness questions. High-worthwhileness questions should enhance interviewees' tendency to flag information items that pragmatically correspond to the specified information objective. And that propensity should manifest irrespective of interviewees' disposition to be cooperative or resistant. Disposition should have a greater influence on what interviewees disclose, and question type should determine what an interviewee perceives an interviewer wants to know.

Contrary to our expectations, the findings on mental designation preferences (i.e. Study 1) suggest that interviewees generally assume interviewers *want to know* complete details, essentially everything, irrespective of question specificity. Most participants flagged complete details as the interviewer's information objective, regardless of whether the interviewer posed specific (i.e. high-worthwhileness) or ambiguous (i.e. low-worthwhileness) questions. Our data were consistent with an almost absent effect of question type. Those findings notwithstanding, there was evidence that the low-worthwhileness question did not increase the interviewees' confidence that their mental designations were what the interviewer wanted to know. That finding suggests that question type plays a role in influencing interviewees' confidence. A replication is needed to provide more conclusive evidence on whether high- versus low-worthwhileness questions lead interviewees to be more confident that they have identified an interviewer's information objectives. Taken together, our findings indicate that intelligence interviewees engage in a reasoning process different from

**Table 5.** Population-level estimates of Model 2 in log-odds with the standard errors and 95% credible intervals. By-designation (grand) intercepts are listed first, followed by the slope estimates for Question-type and Disposition. Slope coefficients that reached the desired level of precision are shown in bold. The slope for Question-type is the change in log-odds for the high-worthwhileness question (1, high-worthwhileness; −1, low-worthwhileness) and the slope for Disposition is the change in log-odds for cooperative participants (1, cooperative; −1, resistant).

| utterance | coefficient | posterior mean | est. error | l-95% CrI | u-95% CrI |
|---|---|---|---|---|---|
| complete details | intercept | 0.04 | 0.38 | −0.72 | 0.76 |
| medium details | intercept | −0.35 | 0.24 | −0.83 | 0.11 |
| no comment | intercept | −1.13 | 0.29 | −1.73 | −0.57 |
| complete details | cooperative/question type: high- versus low- worthwhileness | 0.19 | 0.31 | −0.43 | 0.79 |
| complete details | resistant/question type: high- versus low- worthwhileness | 0.16 | 0.37 | −0.57 | 0.86 |
| medium details | cooperative/question type: high- versus low- worthwhileness | −0.19 | 0.39 | −0.97 | 0.59 |
| medium details | resistant/question type: high- versus low- worthwhileness | 0.17 | 0.25 | −0.34 | 0.65 |
| no comment | cooperative/question-type: high- versus low- worthwhileness | −0.04 | 0.43 | −0.89 | 0.80 |
| no comment | resistant/question type: high- versus low- worthwhileness | 0.18 | 0.22 | −0.25 | 0.62 |
| *complete details* | *disposition: cooperative versus resistant* | *4.26* | *0.49* | *3.29* | *5.22* |
| *medium details* | *disposition: cooperative versus resistant* | *1.66* | *0.29* | *1.07* | *2.23* |
| no comment | disposition: cooperative versus resistant | −1 0.04 | 0.37 | −1.78 | −0.33 |

what we theorized. Interviewees likely consider the interviewer's *overarching* objective goal to acquire *all* relevant information items rather than what an immediate question pragmatically requests.

If there is an effect of question type on the mental designation of information items and interviewees' confidence in those designations, such effect would likely be smaller than we expected. Consider the slope coefficient of Model 1, which examined the effect of high- versus low-worthwhileness questions on mentally designating complete over bare details. That coefficient had a posterior mean of 0.41 and a [0.18,0.65] 95% credible interval. The corresponding predicted difference between the low- and high-worthwhileness questions in influencing interviewees to mentally designate complete details was approximately 12%—averaged over the resistant and cooperative conditions. Put differently, the complete detail designations were 12% more likely in the high-worthwhileness than in the low-worthwhileness condition, which is further illustrated in figure 7.

By contrast, consider a scenario in which the question-type coefficient for complete detail designations in Model 1 is not 0.41 but closer to 0.18 (i.e. the lower 95% CrI bound for the slope coefficient). That hypothetical difference between high- and low-worthwhileness questions would be approximately 4% (averaged over the disposition conditions). It remains a subjective matter whether those differences are something stakeholders should care about when considering the effect of question type on how interviewees hone in on relevant information items. One could argue that an effect of 4% might have meaningful effects in actual interviews. In that case, we suggest a ROPE of [−0.17, 0.17] for future studies examining the effect of high- versus low-worthwhileness questions on the mental designation of information items.

The analysis examining the influence of disposition on mental designations was inconclusive. We did not find support for or against the prediction that a cooperative versus resistant disposition has no

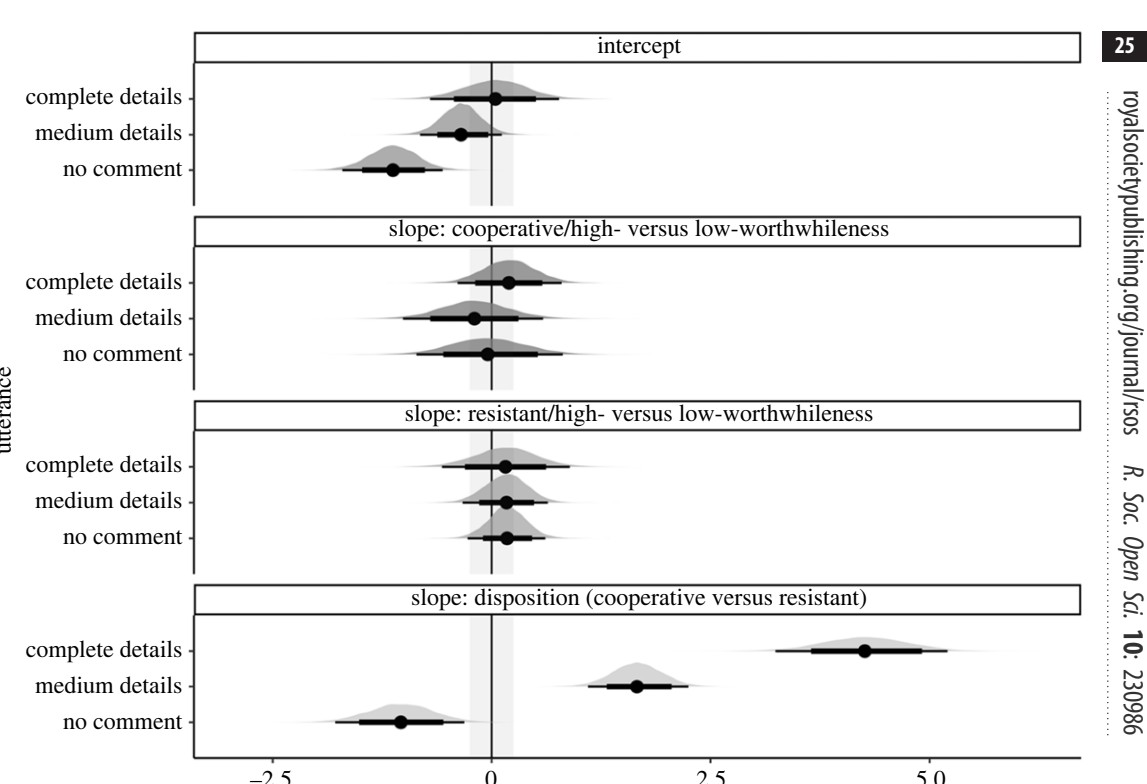

**Figure 6.** Posterior distributions over population-level estimates for Model 2 with 80% and 95% highest density intervals and the ROPE area shaded in light grey.

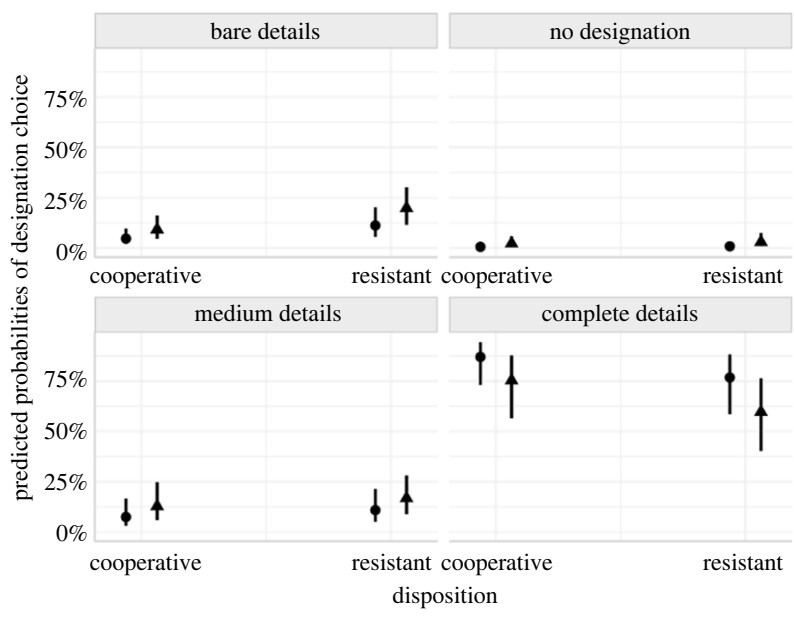

question type ● high-worthwhileness ▲ low-worthwhileness

**Figure 7.** Predicted probabilities of designation choice given the output of Model 1 and our data. The figure contrasts the probability of choosing designations with different levels of details for high- versus low-worthwhileness questions and cooperative versus resistant interviewees.

influence on what interviewees flag as relevant information items. We only found evidence against a theory postulating that resistant versus cooperative interviewees more frequently designate medium details over bare details.

Let us now turn our attention to Study 2, which examined the effect of question type and disposition on what participants disclose. Our goal was to demonstrate that disposition has a greater influence on

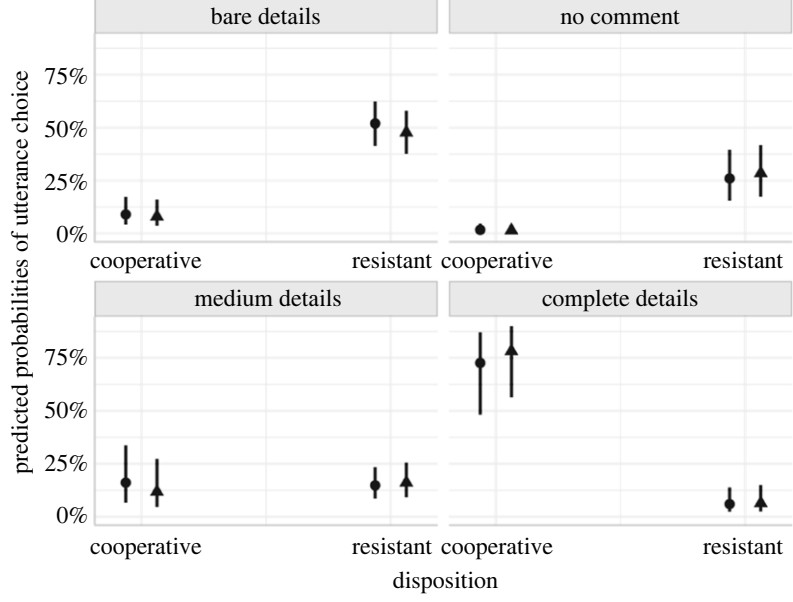

**Figure 8.** Predicted probabilities of utterance choice given the output of Model 2 and our data. The figure contrasts the probability of choosing utterances with different levels of details for high- versus low-worthwhileness questions and cooperative versus resistant interviewees.

disclosure. We did not find any evidence for an influence of question type on the disclosure of cooperative versus resistant interviewees and cannot make any stronger claims due to imprecision. The prediction plot (figure 8) illustrates that there does not seem to be a consistent difference between high- versus low-worthwhileness questions on disclosure.

In line with our hypothesis, there was some evidence that disposition has a stronger influence on interviewees' disclosure than when reasoning about what the interviewer wants to know. We cannot make stronger claims at this point, but we will highlight a few observations. We found a particularly strong difference between resistant and cooperative interviewees in their disclosure of complete details. Averaged over question type, interviewees in the cooperative condition are predicted to be 67% more likely to disclose complete details than those in the resistant condition. Resistant interviewees are predicted to more frequently choose to say nothing at all or disclose bare details (figure 8).

By contrast, cooperative and resistant interviewees are predicted to choose utterances with medium details similarly frequently. This finding, in conjunction with the other findings on utterances, points to an issue worth noting. The disclosure of complete details is likely particular to cooperative interviewees. And the disclosure of bare details or not saying anything is particular to resistant interviewees. However, both cooperative and resistant interviewees seem to refrain from disclosing medium (i.e. partial) details. Given the current findings, it is evident that our decision not to include a semi-cooperative disposition was unwise. Such a disposition could have elicited the disclosure of medium details. We recommend future research to include a semi-cooperative condition to provide a more complete picture of the mechanisms underlying mental designations in comparison to disclosure.

## 8.1. Future directions and concluding remarks

Unlike we predicted, question worthwhileness had no effect on how interviewees determine what an interviewer wants to know in intelligence interviews. It seems interviewees assume their interviewer wants to elicit *everything* on a topic under discussion rather than the specific thing a question requests. Our results suggest that specific or ambiguous questions lead interviewees to mentally flag similar information items (i.e. complete details)—provided those questions are on the same topic.

We speculate that despite the ambiguity of the low-worthwhileness questions, participants may have inferred that the interviewer still wanted to know complete details, given the context of investigations in general. More details would arguably be of greater benefit in an investigation than fewer details. And such context could have influenced participants' perception of the interviewer's objectives. Participants may have assumed the ambiguity during the low-worthwhileness trials was due to the interviewer's

ignorance about the scope of their current discovery. Note that participants underwent both high- and low-worthwhileness trials. The manipulation of *context* in future research could provide insight into how such a variable might influence the effect of high- versus low-worthwhileness questions. For example, it might be useful to examine whether consistently asking low- versus high-worthwhileness questions replicates our findings or reveals different insights.

That notwithstanding, our results suggested that questions with greater specificity (i.e. high- versus low-worthwhileness questions) may increase the confidence in what the interviewer wants to know. We speculate that the confidence high-worthwhileness questions bring might serve two ends. They might facilitate disclosure when interviewees elect to be cooperative. But high-worthwhileness questions might impede disclosure or even assist in deception, given that such questions make interviewees confident in their perception of what the interviewer wants to know. And resistant interviewees are inclined to refrain from assisting their interviewer. Further research is needed to ascertain the speculations just described.

One could argue that our research design could have unduly influenced our results. The objection could be that using multiple-choice response options limited participants' cognitive maps, restricting their potential mental designations and utterances. We disagree. The topics in each scenario were well-defined, making it unlikely for participants to generate mental designations outside each scenario's scope. Interlocutors typically restrict their discourse moves to the topic of discussion unless they wish to ask further questions or change topics [32]. The present study examined the mental designations and utterances elicited by questions under discussion, not how people ask further questions or change topics. Moreover, using multiple-choice options eliminated the possibility of coding errors that can arise from wrangling free-text data.

Our defence notwithstanding, it is possible that the brevity of the respective scenarios limited variability. We acknowledge that our research design may have limited the extent of variation in mental designation preferences and disclosure. Actual events would likely contain a wider range of occurrences than what we included in our scenarios. Given this initial research stage, we opted to prioritize experimental control and internal validity. Future work can build on the ROPE we uncovered. Importantly, our research design provides robust instructions that can be used to guide participants to flag their mental designation choices. Follow-up studies can employ more complex scenarios and free response formats with preregistered codebooks to reduce biased coding of information items. We also recommend the inclusion of a semi-cooperative disposition. Such work will shed more light on the mental designation of information items and allow further theorizing about the implications for intelligence interviewing.

**Data accessibility.** All data supporting the findings in this research are publicly available here: https://osf.io/bgxrj/

**Declaration of AI use.** We have not used AI-assisted technologies in creating this article.

**Authors' contributions.** D.A.N.: conceptualization, data curation, formal analysis, funding acquisition, investigation, methodology, project administration, writing—original draft, writing—review and editing; A.L.: data curation, formal analysis, investigation, methodology, project administration, visualization, writing—review and editing.

Both authors gave final approval for publication and agreed to be held accountable for the work performed therein.

**Conflict of interest declaration.** We declare we have no competing interests.

**Funding.** We received no funding for this study.

# Appendices

## Appendix 1
### Introduction
This study is about communication within a law enforcement context. You will read some fictional scenarios assuming the role of the main character. Then you will answer some questions about each scenario. **Most of this study involves reading. So, please read the scenarios and instructions carefully because understanding them is crucial**. We have included questions to check if you read and answered questions with your full attention.

The entire study will take approximately 10–15 min to complete. You will receive a compensation of £2.25 for participating once the study is over.

### *Appendix A*
### Disposition manipulations
Imagine that you are one of the owners of a restaurant in town; you also work at this restaurant, which overlooks a big park. You and your colleagues have a good picture of what goes on in the park. It is well known among the restaurant staff that a narcotics-dealing gang called KET22 operates in the park. Recently, a police-contact approached you and your colleagues to provide information about the gang if you discovered anything. The police-contact mentioned that none of you are obliged to give any information. (**Dispositional variations begin here**)

**Cooperative:** However, KET22 disrupts your business at the restaurant. So, it is in your best interest to assist the police in their investigations to eliminate the gang. Then your business can grow.

**Resistant:** The police-contact does not know this: but because you (personally) came into some financial troubles, you occasionally supply narcotics to customers at the restaurant on the gang's behalf. If the gang gets busted, you are very likely to get in trouble too. You only agreed to meet with the police-contact to avoid suspicion.

**Manipulation check (disposition)**

Now you know your character or the role you are to play in this study. Suppose you were to make discoveries that could get KET22 busted, and the police-contact asked you about those discoveries. How would you engage with the interviewer?

— I will lie to ensure that I hide what I know. (−1)
— I will keep silent and not respond to the question. (0)
— I will reveal some of my discoveries, not everything I know. (1)
— I will reveal what I know. (2)

*Appendix B*
**Introduction to scenarios**

In the next phase of the study, you will be placed in various scenarios where you will make various discoveries about KET22, the gang under investigation. After each discovery, you will receive a question from the police contact about the discovery. (**Variations between Studies 1 and 2 begin here**)

*Study 1 (Designation experiment):-*

Your task in the upcoming phase is to **indicate what you think the police-contact WANTS TO KNOW about your discovery based on the police-contact's question**. The task is NOT about indicating what you will necessarily say in response to the question.

**Your task is to indicate what you think the police-contact wants to know based on the police-contact's question!**

We have included other questions to check if you read and answered the questions with your full attention.

*Study 2 (Utterance experiment):-*

Your task in the upcoming phase is to **indicate WHAT YOU WANT TO SAY in response to the police-contact's question**.

We have included other questions to check if you read and answered the questions with your full attention.

**IMC (Studies 1 and 2)**

*Passing or failing this IMC will depend on whether a participant undergoes Study 1 or 2.*
What is **TRUE about your main task** in the upcoming phase?

— My task is to indicate what I think the police-contact wants to know based on the police-contact's question.
— My task is to indicate what I want to say in response to the police-contact's question.
Next follows the scenarios.

*Appendix C*

Participants will be randomly assigned to one of two lists to facilitate randomization. And the contents of the lists will be presented in random order in both the utterance and designation conditions.

| List 1 | List 2 |
|---|---|
| Scenario 1a | Scenario 1b |
| Scenario 2b | Scenario 2a |
| Scenario 3a | Scenario 3b |
| Scenario 4b | Scenario 4a |
| Scenario 5a | Scenario 5b |
| Scenario 6b | Scenario 6a |
| Scenario 7a | Scenario 7b |
| Scenario 8b | Scenario 8a |
| Scenario 9a | Scenario 9b |
| Scenario 10b | Scenario 10a |

Before presenting the information items in each scenario, the designation and utterance conditions will include the following prompts, respectively.

— Select what you think the interviewer wants to know. (Study 1, *Designation experiment*)
— Select what you want to say. (Study 2, *Utterance experiment*)

Apart from the information items, the designation and utterance experiments will include the following options, respectively.

— I cannot determine what the interviewer wants to know (Study 1, *Designation experiment*).
— No comment (Study 2, *Utterance experiment*)

Scenarios
(a) High-worthwhileness questions marked with green highlight
(b) Low-worthwhileness questions marked with yellow highlight

1. One day after work, on your bus ride home, you recognized one of the KET22 members. You were sitting just behind him, and he was talking on the phone. He tried to be quiet, but you heard him say: 'It is better to sell the off-brand green-star oxycodone'.

    a. Have you discovered *the particular brand of narcotics* KET22 sells?
    b. Have you discovered anything about the gang's narcotics sales lately?
        i. They sell oxycodone.
        ii. They sell green-star oxycodone.
        iii. They sell the off-brand green-star oxycodone.

2. Your curiosity has led you to pay more attention to KET22's drug dealings. You've discovered that the gangsters usually deal their drugs to customers in the evenings when the workday ends, around 18.00.

    a. Do you have an idea of *what specific time* the gangsters deal drugs to customers?
    b. Do you have any information about when the gang deals drugs?
        i. The drug deals happen in the evening.
        ii. The drug deals happen in the evening when the workday ends.
        iii. The drug deals happen in the evening when the workday ends at 18.00.

3. You always come to work earlier than your colleagues because you supervise the cleaners. You've realized that the KET22 gangsters usually arrive shortly after you in a blue Nissan Qashqai. By paying more attention, you've memorized the license plate number: FBT038.

    a. Do you know the *full details about the vehicle* the KET22 gangsters usually arrive in at the park?
    b. Do you have any information about KET22's transportation in the park?
        i. They usually arrive in a Nissan.
        ii. They usually arrive in a Nissan Qashqai.
        iii. They usually arrive in a Nissan Qashqai, license plate number FBT038.

4. During one of your short breaks at work, you decided to enjoy some sunshine. So, you went to the edge of the park where there are benches. As you approached, you saw a rowdy group at one of the benches, and you chose the bench furthest away from them. The group was talking about how to contact KET22 to buy narcotics. They said customers could make contact by sending a text message containing a lion emoji to any KET22 phone number.

    a. Have you made observations about *exactly how customers contact* KET22 to buy narcotics?
    b. Have you made any observations about KET22's customers?
        i. Customers make contact by phone.
        ii. Customers make contact by sending a text message.
        iii. Customers make contact by sending a text message containing a lion emoji.

5. Lately, you have noticed a particular spot at the park where the KET22 gangsters deal drugs. The spot is one of the park's exits, EXIT 7F. All the exits are located at different edges of the park, but 7F is rather discreet.

    a. Have you spotted *the exact location* at the park where KET22 deals drugs?
    b. Have you spotted anything about where KET22 deals drugs?
        i. The gangsters deal at the edge of the park.
        ii. The gangsters deal at an exit at the edge of the park.
        iii. The gangsters deal at EXIT 7F, a discreet location at the edge of the park.

6. From your observations of the park, it seems that the gangsters take shifts selling their narcotics. Their rotations include five (5) gangsters, two (2) sell, and three (3) are lookouts.

    a. Have you discovered the specific methods the gangsters use when selling narcotics?

b. Have you observed anything about the gang's drug sales?

 i. They take shifts selling narcotics.

 ii. They take shifts selling narcotics, five gangsters at a time.

 iii. They take shifts selling narcotics, five (5) gangsters at a time; two (2) sell, and three (3) are lookouts.

7. On your way home after work, you saw that some KET22 gangsters were arguing. It was around 19.00 on Monday. From what you heard, the argument was about whether to sell a high dose of drugs to a customer.

 a. Have you caught *the contents of particular interactions* between the gang members lately?

 b. Have there been any developments with the gang members lately?

 i. There was an argument between some gang members.

 ii. On Monday at 19.00, there was an argument between some gang members.

 iii. On Monday at 19.00, there was an argument between some gang members about whether to sell a high dose of drugs to a customer.

8. Your colleague, who is becoming friends with a KET22 gangster, recently slipped you some details. She said that KET22 is connected to a much bigger gang called TETO. TETO supplies opioids wholesale.

 a. Do you have information about the sources from which KET22 obtains narcotics?

 b. Has anything about KET22's narcotics operations come to your attention?

 i. KET22 is connected to a much bigger gang.

 ii. KET22 is connected to a much bigger gang called TETO.

 iii. KET22 is connected to a much bigger gang called TETO that supplies opioids wholesale.

9. KET22 has now added security measures to hide from the police: You discovered this because, after work, you decided to walk home through the woods. You overhead some voices in a secluded area and decided to get a better look. When you got close enough, you recognized that it was some KET22 gangsters. You saw that they were trying to set up some radio communication systems. One of the gangsters was explaining that the communication systems would be used to send alerts about police presence.

 a. Have you come across the particular strategies the gang uses to avoid police detection?

 b. Have you observed anything about the gang's recent activities?

 i. They have adopted security measures to hide from the police.

 ii. They have adopted radio communication systems as a security measure to hide from the police.

 iii. They have adopted radio communication systems as a security measure to hide from the police; they use the radios to send alerts about police presence.

10. You've noticed that a pickup truck makes deliveries to the KET22 gangsters in the park. The deliveries usually come before the restaurant opens.

 a. Do you know the specific ways the gangsters get deliveries in the park?

 b. Do you know anything about the gang's drug operations in the park?

 i. The gangsters use a pickup truck in the park.

 ii. A pickup truck makes deliveries to the gangsters in the park.

 iii. A pickup truck makes deliveries to the gangsters in the park before the restaurant opens.

**Confidence rating (After each scenario in Study 1 [Designation experiment])**

— The police-contact asked:

 [*display question*]

— Based on the above question, you selected the option below as what the police-contact wants to know:

 [*display selection*]

 — On a scale from 1 to 5, how confident are you that *your selection* is what the police-contact wants to know?

 — 1 = *not confident at all*, 2 = *slightly confident*, 3 = *somewhat confident*, 4 = *fairly confident*, 5 = *completely confident*

— **Confidence measure via bets [optional question]**

 ■ Suppose you were to place a bet on your selection. On a scale from 0 to 100, what percentage of your compensation (for participating in this research) are you willing to bet that *your selection* is what the police-contact wants to know.

 ■ 0% = none of my compensation, 100% = all of my compensation

*Appendix D*
## Control Questions

*The control questions will employ the same scenario outlined below. The scenario will be presented four times, in random order, with four different questions.*

Recently, a man came into the restaurant to buy coffee. You suspect he might be one of the KET22 gangsters, but you are unsure. When he made his order, there was no milk at the counter. So, you asked your colleague to get some milk from the fridge in the back. While you were waiting, you got a good look at his face and stature. You can guess that he is about 190 cm tall. His hair was dark with grey streaks. He had green eyes and a scar on his left jaw. The name on the card he used to pay for his drink was Kari Jupo.

Q1. From the options below, select the name on the card the man used to pay for his drink.

— Minea Blankson
— Johnny Biles
— Kari Jupo
— Renave Olsson

Q2. From the options below, select the correct description of the man's hair.

— Blonde with brown streaks
— Blonde with grey streaks
— Dark with grey streaks
— Dark with yellow streaks

Q3. From the options below, select what the man ordered.

— Sandwich
— Coffee
— Beer
— Salad

Q4. From the options below, select the correct description of the man's height.

— 190 cm
— 200 cm
— 164 cm
— 175 cm

## Appendix III: Interaction Model 1

| designation | coefficient | posterior mean | est. error | l-95% CrI | u-95% CrI |
|---|---|---|---|---|---|
| complete details | intercept | 2.02 | 0.35 | 1.32 | 2.78 |
| medium details | intercept | 0.16 | 0.18 | −0.20 | 0.53 |
| no designation | intercept | −2.09 | 0.38 | −2.91 | −1.40 |
| complete details | question type: high- versus low-worthwhileness | 0.43 | 0.12 | 0.20 | 0.68 |
| medium details | question type: high- versus low-worthwhileness | 0.09 | 0.13 | −0.18 | 0.35 |
| no designation | question type: high- versus low-worthwhileness | −0.34 | 0.25 | −0.84 | 0.18 |
| complete details | disposition: cooperative versus resistant | 0.52 | 0.22 | 0.08 | 0.95 |
| medium details | disposition: cooperative versus resistant | 0.26 | 0.11 | 0.04 | 0.49 |
| no designation | disposition: cooperative versus resistant | 0.35 | 0.23 | −0.08 | 0.81 |
| complete details | cooperative: high- versus low-worthwhileness | 0.07 | 0.10 | −0.14 | 0.27 |
| medium details | cooperative: high- versus low-worthwhileness | 0.09 | 0.09 | −0.10 | 0.27 |
| no designation | cooperative: high- versus low-worthwhileness | 0.37 | 0.22 | −0.06 | 0.82 |

## Appendix IV: Prior sensitivity

Model 1 with the following priors:

  Intercept: N(0, 10)
  Fixed effects: N(0, 1)
  Random effects s.d.: Half Normal(0, 1)
  Correlation: LKJ(2)

| designation | coefficient | posterior mean | est. error | l-95% Crl | u-95% Crl |
| --- | --- | --- | --- | --- | --- |
| complete details | intercept | 2.03 | 0.35 | 1.33 | 2.73 |
| medium details | intercept | 0.15 | 0.18 | −0.21 | 0.51 |
| no designation | intercept | −2.02 | 0.37 | −2.80 | −1.34 |
| complete details | question type: high- versus low-worthwhileness | 0.41 | 0.12 | 0.18 | 0.65 |
| medium details | question type: high- versus low-worthwhileness | 0.07 | 0.13 | −0.20 | 0.32 |
| no designation | question type: high- versus low-worthwhileness | −0.33 | 0.25 | −0.81 | 0.18 |
| complete details | disposition:cooperative versus resistant | 0.50 | 0.21 | 0.09 | 0.92 |
| medium details | disposition:cooperative versus resistant | 0.25 | 0.11 | 0.03 | 0.48 |
| no designation | disposition: cooperative versus resistant | 0.25 | 0.22 | −0.16 | 0.70 |

Model 1 with the following priors:

  Intercept: student_t(3, 0, 2.5)
  Fixed effects: flat prior
  Random effects sd: student_t(3, 0, 2.5)
  Correlation: LKJ(1)

| designation | coefficient | posterior mean | est. error | l-95% Crl | u-95% Crl |
| --- | --- | --- | --- | --- | --- |
| complete details | intercept | 2.02 | 0.38 | 1.27 | 2.77 |
| medium details | intercept | 0.16 | 0.19 | −0.22 | 0.52 |
| no designation | intercept | −2.02 | 0.39 | −2.87 | −1.31 |
| complete details | question type: high- versus low-worthwhileness | 0.42 | 0.13 | 0.17 | 0.69 |
| medium details | question type: high- versus low-worthwhileness | 0.07 | 0.13 | −0.19 | 0.33 |
| no designation | question type: high- versus low-worthwhileness | −0.32 | 0.29 | −0.90 | 0.26 |
| complete details | disposition: cooperative versus resistant | 0.54 | 0.23 | 0.08 | 0.99 |
| medium details | disposition: cooperative versus resistant | 0.26 | 0.12 | 0.03 | 0.49 |
| no designation | disposition: cooperative versus resistant | 0.28 | 0.24 | −0.16 | 0.78 |

Model 2 with the following priors:

Intercept: N(0, 10)
Fixed effects: N(0, 1)
Random effects sd: Half Normal(0, 1)
Correlation: LKJ(2)

| utterance | coefficient | posterior mean | est. error | l-95% CrI | u-95% CrI |
|---|---|---|---|---|---|
| complete details | intercept | 0.03 | 0.37 | −0.70 | 0.76 |
| medium details | intercept | −0.35 | 0.23 | −0.82 | 0.10 |
| no comment | intercept | −1.14 | 0.29 | −1.75 | −0.59 |
| complete details | cooperative/question type: high- versus low-worthwhileness | 0.19 | 0.31 | −0.42 | 0.80 |
| complete details | resistant/question type: high- versus low-worthwhileness | 0.16 | 0.37 | −0.58 | 0.87 |
| medium details | cooperative/question type: high- versus low-worthwhileness | −0.20 | 0.40 | −0.97 | 0.59 |
| medium details | resistant/question type: high- versus low-worthwhileness | 0.17 | 0.25 | −0.35 | 0.65 |
| no comment | cooperative/question type: high- versus low-worthwhileness | −0.04 | 0.42 | −0.87 | 0.80 |
| no comment | resistant/question type: high- versus low-worthwhileness | 0.18 | 0.22 | −0.25 | 0.63 |
| complete details | disposition: cooperative versus resistant | 4.26 | 0.50 | 3.27 | 5.23 |
| medium details | disposition: cooperative versus resistant | 1.65 | 0.29 | 1.09 | 2.23 |
| no comment | disposition: cooperative versus resistant | −1 0.05 | 0.38 | −1.78 | −0.31 |

Model 2 with the following priors:

Intercept: student_t(3, 0, 2.5)
Fixed effects: flat prior
Random effects sd: student_t(3, 0, 2.5)
Correlation: LKJ(1)

| utterance | coefficient | posterior mean | est. error | l-95% CrI | u-95% CrI |
|---|---|---|---|---|---|
| complete details | intercept | −0.01 | 0.43 | −0.87 | 0.84 |
| medium details | intercept | −0.30 | 0.26 | −0.84 | 0.20 |
| no comment | intercept | −1.15 | 0.32 | −1.79 | −0.54 |
| complete details | cooperative/question type: high- versus low-worthwhileness | 0.20 | 0.39 | −0.58 | 0.96 |
| complete details | resistant/question type: high- versus low-worthwhileness | 0.25 | 0.52 | −0.80 | 1.30 |
| medium details | cooperative/question type: high- versus low-worthwhileness | −0.22 | 0.50 | −1.23 | 0.78 |
| medium details | resistant/question type: high- versus low-worthwhileness | 0.18 | 0.29 | −0.40 | 0.73 |
| no comment | cooperative/question type: high- versus low-worthwhileness | −0.01 | 0.56 | −1.11 | 1.09 |
| no comment | resistant/question type: high- versus low-worthwhileness | 0.21 | 0.25 | −0.27 | 0.71 |
| complete details | disposition: cooperative versus resistant | 6.09 | 0.72 | 4.77 | 7.57 |
| medium details | disposition: cooperative versus resistant | 1.94 | 0.32 | 1.32 | 2.59 |
| no comment | disposition: cooperative versus resistant | −1 0.14 | 0.44 | −2.04 | −0.30 |

Model 3 with the following priors:

Intercept: N(0, 10)
Fixed effects: N(0, 1)
Random effects sd: Half Normal(0, 1)
Correlation: LKJ(2)

| estimate | posterior mean | est. error | l-95% CrI | u-95% CrI |
|---|---|---|---|---|
| Intercept1 | −3.55 | 0.19 | −3.94 | −3.18 |
| Intercept2 | −2.44 | 0.15 | −2.73 | −2.15 |
| Intercept3 | −1.35 | 0.14 | −1.62 | −1.08 |
| Intercept4 | 0.24 | 0.13 | −0.02 | 0.50 |
| question type: high- versus low-worthwhileness | 0.16 | 0.05 | 0.07 | 0.25 |
| disposition: cooperative versus resistant | 0.11 | 0.10 | −0.08 | 0.29 |

Model 3 with the following priors:

  Intercept: student_t(3, 0, 2.5)
  Fixed effects: flat prior
  Random effects sd: student_t(3, 0, 2.5)
  Correlation: LKJ(1)

| estimate | posterior mean | est. error | l-95% Crl | u-95% Crl |
| --- | --- | --- | --- | --- |
| Intercept1 | −3.53 | 0.19 | −3.90 | −3.16 |
| Intercept2 | −2.41 | 0.15 | −2.71 | −2.12 |
| Intercept3 | −1.33 | 0.14 | −1.60 | −1.05 |
| Intercept4 | 0.26 | 0.14 | −0.01 | 0.53 |
| question type: high- versus low-worthwhileness | 0.15 | 0.05 | 0.06 | 0.24 |
| disposition: cooperative versus resistant | 0.11 | 0.10 | −0.09 | 0.29 |

Model 4 with the following priors:
  Intercept: N(0, 10)
  Fixed effects: N(0, 1)
  Random effects sd: Half Normal(0, 1)
  Correlation: LKJ(2)

| estimate | posterior mean | est. error | l-95% Crl | u-95% Crl |
| --- | --- | --- | --- | --- |
| intercept | 4.41 | 0.44 | 3.58 | 5.31 |
| question type: high- versus low-worthwhileness | 0.49 | 0.21 | 0.08 | 0.94 |
| disposition: cooperative versus resistant | 0.18 | 0.26 | −0.33 | 0.67 |

Model 4 with the following priors:

  Intercept: student_t(3, 0, 2.5)
  Fixed effects: flat prior
  Random effects sd: student_t(3, 0, 2.5)
  Correlation: LKJ(1)

| estimate | posterior mean | est. error | l-95% Crl | u-95% Crl |
| --- | --- | --- | --- | --- |
| intercept | 4.57 | 0.51 | 3.62 | 5.61 |
| question type: high- versus low-worthwhileness | 0.55 | 0.27 | 0.04 | 1.14 |
| disposition: cooperative versus resistant | 0.22 | 0.29 | −0.36 | 0.79 |

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
