## [Peer Review File · Royal Society Open Science]

Review History

Decision letter (RSOS-230986.R0)

Dear Dr Neequaye

On behalf of the Editor, I am pleased to inform you that your Stage 1 Manuscript RSOS-230986 entitled "How Intelligence Interviewees Mentally Identify Relevant Information" has been accepted in principle for publication in Royal Society Open Science.

We will now process the Stage 2 submission on your behalf. Please note there is no action required from you at this stage.

If you have any questions at all, please do not hesitate to get in touch.

on behalf of Professor Chris Chambers (Associate Editor) and Chris Chambers (Registered Reports Editor, Royal Society Open Science)
openscience@royalsociety.org

Author's Response to Decision Letter for (RSOS-230986.R0)

See Appendix A.

RSOS-230986.R1 (Revision)

Decision letter (RSOS-230986.R1)

Dear Dr Neequaye:

I am pleased to inform you that your manuscript entitled "How Intelligence Interviewees Mentally Identify Relevant Information" is now accepted for publication in Royal Society Open Science.

Please remember to make any data sets or code libraries 'live' prior to publication, and update any links as needed when you receive a proof to check - for instance, from a private 'for review' URL to a publicly accessible 'for publication' URL. It is also good practice to add data sets, code and other digital materials to your reference list.

Royal Society Open Science is a fully open access journal. A payment may be due before your article is published. Please note that, if the corresponding author of your paper is based at an institution covered by one of our Transformative Agreement deals, your fees may be covered by the deal - please check the list of eligible institutions at <https://royalsociety.org/journals/authors/read-and-publish/read-publish-agreements/>. The Royal Society has partnered with Copyright Clearance Center's (CCC's) RightsLink service to allow authors to pay article processing charges or page charges. After your manuscript has been accepted, the corresponding author will receive an email from CCC with the subject "Please submit your article processing/open access charge(s)/page charges" inviting you to pay your charges or request an invoice. The email from CCC will come from the email domain @copyright.com (if you have any queries regarding fees, please see <https://royalsocietypublishing.org/rsos/charges> or contact authorfees@royalsociety.org). If you request an invoice, it will be sent to you from CCC. It is important to be cautious about payment scams.

If you receive an email or text message requesting payment and have any concerns, we recommend contacting us through our website, rather than clicking on any links. The Royal Society will never ask you to make a direct payment.

Your feedback matters - please spend 5 minutes leaving anonymous feedback about your experience of Registered Reports at this journal, as an author or reviewer:
https://registeredreports.cardiff.ac.uk/feedback/feedback/decision_letter.php

This feedback is collected by the Registered Reports Community Feedback website, which is an independent service and research project, being undertaken by Cardiff University.

on behalf of Professor Professor Chris Chambers (Subject Editor).

Professor Chris Chambers Comments to Author:

In the proofs, at various stages please replace the term "PCI Registered Report" with "PCI Registered Reports".

<https://www.facebook.com/RoyalSocietyPublishing/>

Appendix A

This research is a Registered Report reviewed and recommended by PCI Registered Report.

- Link to Stage 1 Recommendation:
<https://rr.peercommunityin.org/articles/rec?id=188>
- Link to Stage 2 Recommendation:
<https://rr.peercommunityin.org/articles/rec?id=476>

This Registered Report was submitted to Royal Society Open Science following peer review and recommendation for Stage 2 acceptance at the Peer Community In (PCI) Registered Reports platform. Full details of the peer review and recommendation of the paper at PCI Registered Reports may be found at the links below.

After submission to the journal, the paper received no additional external peer review, but was accepted on the basis of the Editor's recommendation according to our PCI Registered Reports policy <https://royalsocietypublishing.org/rsos/registered-reports#PCIRR>.